The EMBO Journal (2013) 32, 2172–2185
www.embojournal.org

# DNA polymerase κ-dependent DNA synthesis at stalled replication forks is important for CHK1 activation

Rémy Bétous[1,2,6],
Marie-Jeanne Pillaire[1,2,6], Laura Pierini[1,2],
Siem van der Laan[3], Bénédicte Recolin[3,7],
Emma Ohl-Séguy[1,2], Caixia Guo[4],
Naoko Niimi[5], Petr Grúz[5],
Takehiko Nohmi[5], Errol Friedberg[4],
Christophe Cazaux[1,2],
Domenico Maiorano[3,*]
and Jean-Sébastien Hoffmann[1,2,*]

[1]Equipe Labellisée La Ligue Contre le Cancer 2013, INSERM UMR 1037, CNRS ERL 505294, CRCT (Cancer Research Center of Toulouse), Toulouse, France, [2]University of Toulouse; UPS, Toulouse, France, [3]Institut de Génétique Humaine (IGH), CNRS—UPR 1142, Montpellier Cedex 5, France, [4]Department of Pathology, The University of Texas Southwestern Medical Center, Dallas, TX, USA and [5]Division of Genetics and Mutagenesis, National Institute of Health Sciences, Tokyo, Japan

**Formation of primed single-stranded DNA at stalled replication forks triggers activation of the replication checkpoint signalling cascade resulting in the ATR-mediated phosphorylation of the Chk1 protein kinase, thus preventing genomic instability. By using siRNA-mediated depletion in human cells and immunodepletion and reconstitution experiments in *Xenopus* egg extracts, we report that the Y-family translesion (TLS) DNA polymerase kappa (Pol κ) contributes to the replication checkpoint response and is required for recovery after replication stress. We found that Pol κ is implicated in the synthesis of short DNA intermediates at stalled forks, facilitating the recruitment of the 9-1-1 checkpoint clamp. Furthermore, we show that Pol κ interacts with the Rad9 subunit of the 9-1-1 complex. Finally, we show that this novel checkpoint function of Pol κ is required for the maintenance of genomic stability and cell proliferation in unstressed human cells.**

*The EMBO Journal* (2013) **32,** 2172–2185. doi:10.1038/emboj.2013.148; Published online 25 June 2013
*Subject Categories:* genome stability & dynamics
*Keywords*: DNA polymerase κ; genetic instability; replication checkpoint; replication stress

*Corresponding author. D Maiorano, Institut de Génétique Humaine (IGH), CNRS—UPR 1142, 141 rue de la Cardonille, 34396 Montpellier Cedex 5, France. Tel.: + 33 4 34 35 99 73; Fax: + 33 4 34 35 99 01; E-mail: domenico.maiorano@igh.cnrs.fr or
J-S Hoffmann, Equipe Labellisée La Ligue Contre le Cancer 2013, INSERM UMR 1037, CNRS ERL 505294, CRCT (Cancer Research Centre 06 Toulouse)Toulouse 31077, France.
Tel.: + 33 5 61 17 59 75; Fax: + 33 5 61 17 59 94; E-mail: jseb@ipbs.fr
[6]These authors contributed equally to this work.
[7]Present address: Department of Experimental Oncology, University Medical Center Utrecht, Stratenum 2.241, Universiteitsweg 100, 3584 CG Utrecht, The Netherlands.

## Introduction

Common impediments to replication fork progression include exogenous and endogenous DNA damage, natural structured DNA, or tightly associated DNA–protein complexes that disturb the progression of the replicative DNA polymerases α, δ and ε. Failure to stabilize and restart stalled forks or prolonged arrest of replication forks (replication stress) may result in fork collapse, leading to chromosomal breakage and rearrangement. Cells have evolved several mechanisms to deal with the constant challenge of replication stress. These include systems to detect and repair damaged DNA, and replication checkpoints that sense stalled replication forks in S phase to direct appropriate cellular responses (Nyberg *et al*, 2002; Ciccia and Elledge, 2010). When these checkpoints are defective, cells enter cell division with uncompleted DNA replication, and as a consequence genetic instability is increased.

The ATR signalling pathway is crucial in regulating the replication stress response to a large array of insults such as DNA damaging agents and chemicals that cause replication arrest (Melo and Toczyski, 2002; Cimprich and Cortez, 2008). Many of these types of replication arrest cause functional uncoupling of the MCM helicase and replicative DNA polymerase activities at replication forks, resulting in production of long stretches of RPA-coated single-stranded DNA (ssDNA) (Byun *et al*, 2005; Cortez, 2005). In turn, accumulation of the major ssDNA binding protein RPA onto this substrate is thought to trigger the primary checkpoint signalling (Zou *et al*, 2003), although more recent data suggest that the role of RPA in this process may be indirect, by allowing formation of replication forks and very likely by stimulating DNA polymerase α (Pol α)-dependent synthesis of replication intermediates that strongly contribute to checkpoint activation (Van *et al*, 2010; Recolin *et al*, 2012 and see below). The ATR checkpoint pathway is known to involve the independent translocation of multicomponent protein complexes to damage sites followed by the phosphorylation of the main ATR effector, the protein kinase Chk1 (Zou *et al*, 2002; Cimprich and Cortez, 2008). ATR activation requires its localization at stalled forks with its partner ATRIP (ATR Interacting Protein) as well as the heterotrimeric 9-1-1 complex and the ATR activator TopBP1, which are recruited independently of ATR-ATRIP. TopBP1 interacts with both the 9-1-1 complex and ATR-ATRIP complexes, and it plays a role in loading and/or stabilizing the 9-1-1 complex on damaged chromatin (Delacroix *et al*, 2007; Mordes *et al*, 2008; Yan and Michael, 2009; Gong *et al*, 2010). An alternative RFC-containing Rad17 complex loads the 9-1-1 complex to primers synthetized by DNA polymerase alpha (Pol α) (Zou *et al*, 2002; Ellison and Stillman, 2003) and the interaction between Rad9 and TopBP1 positions TopBP1 for ATR activation (Furuya *et al*, 2004; Delacroix *et al*, 2007). In addition, the regulatory protein Claspin has been shown to be required for ATR-dependent phosphorylation of Chk1

(Liu *et al*, 2006). Previous studies have shown that DNA synthesis also is required for checkpoint activation (Michael *et al*, 2000; Lupardus *et al*, 2002). A recent study has shown that continued synthesis and elongation of new DNA primers on ssDNA templates represent an additional key step in checkpoint activation (Van *et al*, 2010).

An important part of the cellular response to replication arrest or stalling by DNA damage is the induction of DNA damage tolerance pathways. The human Y-family DNA polymerases (Pol η, Pol κ, Pol ι, and Rev1) participate in these pathways by facilitating the replicative bypass of DNA lesions. Y-family DNA polymerases promote damage tolerance in part through their ability to insert nucleotides opposite to DNA lesions that block the replicative DNA polymerases, a process termed as translesion synthesis (TLS) (Friedberg *et al*, 2002). Translesion polymerases may function directly at the replication fork (Friedberg, 2005), or may fill in postreplication gaps containing lesions that are left behind by replication forks (Lopes *et al*, 2006; Callegari *et al*, 2010). Ubiquitylation of PCNA, which is dependent upon the E2 ubiquitin-conjugating enzyme Rad6 and the E3 ubiquitin ligase Rad18, facilitates these processes (Friedberg *et al*, 2005). Pol κ, an 870 amino-acid residue Y-family polymerase related to DNA polymerase IV of *Escherichia coli*, can replicate past several bulky adducts in DNA *in vitro*, and cellular studies have shown a role for Pol κ in TLS past lesions such as benzo(a)pyrene-DNA adducts (Ogi *et al*, 2002). However, Pol κ differs from the three other Y-family human DNA polymerases in that mice lacking Pol κ exhibit, besides a spontaneous mutator phenotype which could be explained by defective TLS across endogenously generated adducts (Stancel *et al*, 2009), increased tandem repeat mutations (Burr *et al*, 2006) suggesting that Pol κ could have other roles in addition to TLS. An important, as yet unsolved question is how TLS and replication checkpoint activities are coordinated at stalled replication forks to maintain genomic stability while ensuring the resumption of DNA replication. In the yeast *S. pombe*, it was previously observed that DinB, an orthologue of Pol κ, and the Rad1 and Hus1 components of the 9-1-1 complex interact, suggesting a role for Pol κ in checkpoint responses although the significance of this interaction has not been further explored (Kai and Wang, 2003). We have investigated this issue in both *Xenopus* extracts and mammalian cells, and provide evidence that Pol κ has an additional TLS-independent function in replication checkpoint activation when the replication fork progression is impeded by either nucleotide starvation or upon inhibition of the replicative polymerases. We show that Pol κ fulfils this novel checkpoint function by participating in DNA synthesis on ssDNA at stalled replication forks. We also show that Pol κ is required in otherwise unstressed human cells for normal S-phase progression, nuclear replication factory maturation and the maintenance of genomic stability by preventing the persistence of under-replicated regions in mitosis.

## Results

### Activation of the replication checkpoint is deficient following Pol κ depletion

Short-interfering (si) RNA-mediated *Pol K* gene silencing was achieved in the human U2OS, MRC5 and HeLa cell lines, with substantial depletion of Pol κ (80–90% depletion) by four independent siRNAs or a pool of siRNAs (Figure 1A and B; Supplementary Figure S1A and B). Significant RNAi-mediated depletion of Pol κ was achieved without significantly altering the expression of other replicative or Y-family TLS DNA polymerases (Supplementary Figure S1C). We then assayed Chk1 phosphorylation on serine 345 in Pol κ-depleted human cells in response to nucleotide starvation by Hydroxyurea (HU), a strong inducer of replication stress widely used to investigate responses to DNA damage-independent replication fork arrest (Lopes *et al*, 2001; Sogo *et al*, 2002). We observed a strong and reproducible attenuation of Chk1 phosphorylation when Pol κ was downregulated in several human cell lines (U20S, MRC5 and 293T cells) treated with HU as compared with their counterparts Pol κ-proficient cells (Figure 1A and B; Supplementary Figure S1D). As can be seen in Supplementary Figure S1E, the level of neither ATR, ATRIP, Rad9, Hus1, Pol δ nor TopBP1 was affected following Pol κ depletion, supporting that defective Chk1 phosphorylation is not due to the downregulation of these essential checkpoint factors. Importantly, expression of wild-type Pol κ in siRNA-mediated Pol κ-depleted cells significantly rescued defective Chk1 phosphorylation (Figure 1C). Defective Chk1 phosphorylation was reproduced in Pol κ-knockout primary Murine Embryonic Fibroblast (MEF) cells (Supplementary Figure S1F), and was not observed upon knockdown of Pol η, another Y-family translesion polymerase member (Figure 1D; Bergoglio *et al*, 2013), demonstrating that this phenotype is Pol κ specific. Furthermore, while ectopic expression of wild-type Pol κ did not change the level of Chk1 phosphorylation after HU (Supplementary Figure S1G), expression of catalytically inactive Pol κ (DEAD Pol κ) in which two critical active site residues (D198A and E199A) in the catalytically active site had been mutated (Gerlach *et al*, 2001) mimics the effect of *POL K* gene silencing (Figure 1E), suggesting that Pol κ catalytic activity may be important for checkpoint activation. We also found that Pol κ is recruited to chromatin after HU treatment, underlining a requirement of Pol κ in the response to fork stalling (Supplementary Figure S2A).

In order to obtain more insight in the molecular mechanism by which Pol κ promotes Chk1 phosphorylation, we used extracts derived from activated *Xenopus* eggs naturally synchronized in S-phase. This system recapitulates *in vitro* the regulated activation of the replication checkpoint upon addition of demembranated sperm chromatin, and is amenable to biochemical manipulation (Costanzo and Gautier, 2004). We first generated *Xenopus* Pol κ (XPol κ)-specific antibodies (see Materials and methods and Supplementary Figure S3A) to deplete XPol κ from egg extracts (Figure 2A). We then assessed checkpoint activation after stalling replication forks with either aphidicolin or UV irradiation. Aphidicolin induces fork stalling like HU by inhibiting the activity of replicative DNA polymerases, while UV-induced lesions physically stall replicative DNA polymerases and induce both strong replication fork uncoupling and Chk1 phosphorylation (Byun *et al*, 2005). We observed a strong reduction in Chk1 phosphorylation upon removal of XPol κ, either after aphidicolin (Figure 2B) or UV treatment (Supplementary Figure S3B), as observed in mammalian cells (Figure 1; Supplementary Figure S1D and S2B). Addition of purified recombinant Pol κ to egg extracts

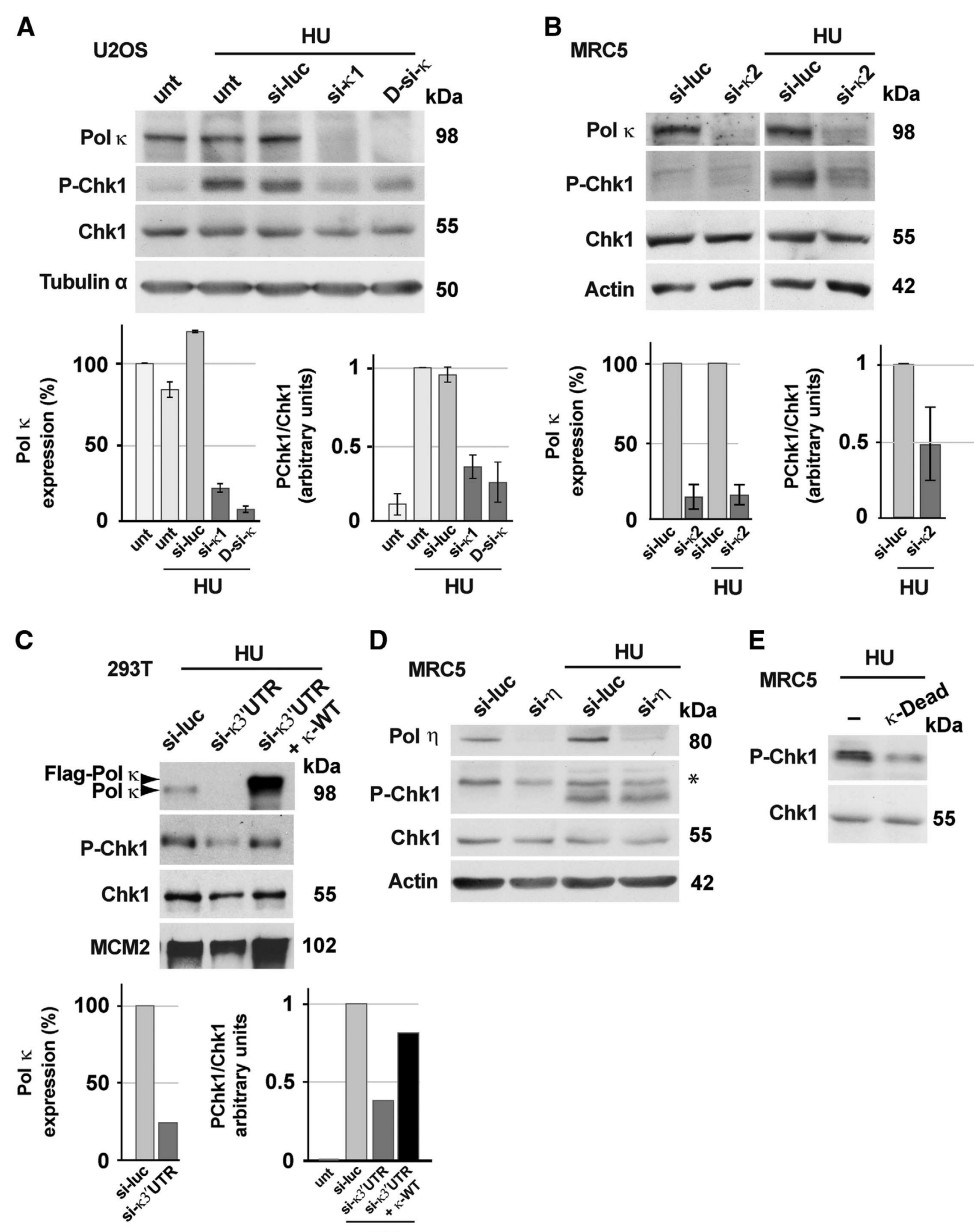

**Figure 1** Pol κ is required for Chk1 phosphorylation after replication stress in human cells. (**A**, **B**) Western blot analysis of cell extracts from Human U2OS and MRC5 untransfected (unt) or transfected with control luciferase siRNA (si-luc), Pol κ siRNA (si-κ1, si-κ2, D-si-κ), untreated or treated with HU (2 mM, 3 h). Cell extracts were fractionated and soluble fractions were analysed with the indicated antibodies. Quantification was performed by ImageJ software; means ± s.d. were presented. (**C**) Western blot analysis of whole cell extracts prepared from 293T cells treated with HU (2 mM, 3 h) transfected by si-RNA alone (si-luc or si-κ3′UTR) or co-transfected by si-κ3′UTR and a vector expressing FLAG-tagged wild-type Pol κ (κ-WT). (**D**, **E**) Pol η depletion with Pol η siRNA (si-η) or transfection with vector expressing the FLAG-tagged catalytic inactive mutant of Pol κ (Pol κ-Dead) was performed in MRC5 cells untreated or treated with HU (2 mM, 3 h). Source data for this figure is available on the online supplementary information page.

depleted of XPol κ totally rescued defective Chk1 phosphorylation (Figure 2B), demonstrating that Pol κ on its own rather than co-immunodepleted proteins is responsible for checkpoint activation, further demonstrating the specificity of Pol κ in this process. Since DNA synthesis is essential for establishment of the S-phase checkpoint (Michael *et al*, 2000; Lupardus *et al*, 2002; Recolin *et al*, 2012), we next verified that DNA replication occurs upon removal of Xpol κ (Figure 2D), demonstrating that Xpol κ is not required for bulk chromosomal replication, and excluding an indirect effect on replication forks formation.

We also obtained independent evidence that *Xenopus* Pol κ-immunodepleted extracts display a deficient ATR/Chk1 signalling pathway by monitoring replication of chromatin containing UV blocking lesions which leads to DNA synthesis slow down (Lupardus *et al*, 2002; Byun *et al*, 2005). As can be seen in Figure 2C, the slow-down of DNA synthesis observed upon UV irradiation of sperm chromatin in control extracts (ΔMock) could be alleviated by addition of caffeine, an ATR/ATM inhibitor. In contrast, extracts lacking XPol κ (ΔXPol κ) displayed a high degree of UV-resistant DNA synthesis that was not enhanced by addition of caffeine (Figure 2C),

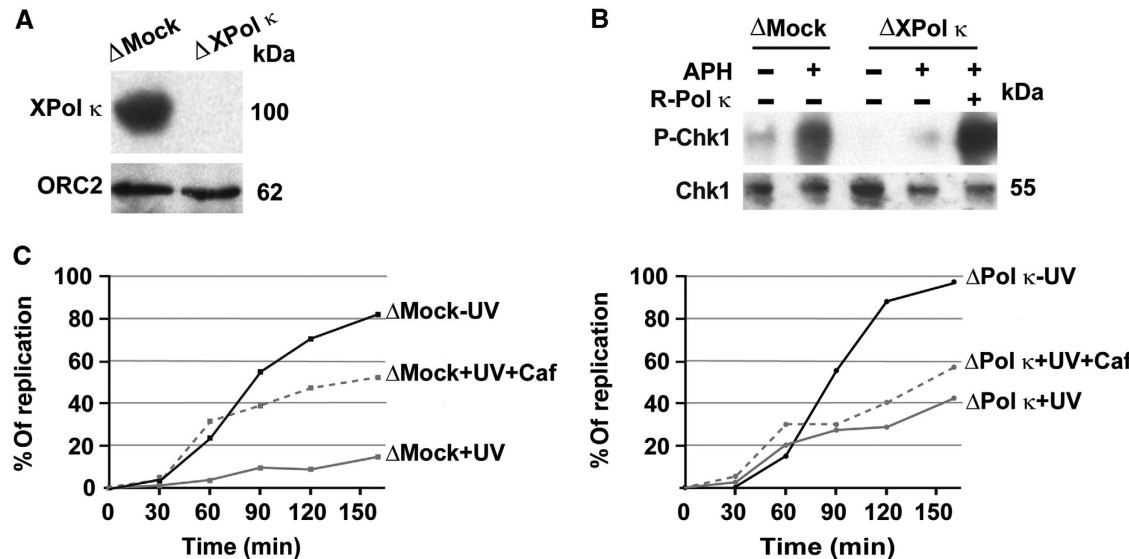

**Figure 2** Pol κ depletion from Xenopus egg extracts affects Chk1 phosphorylation. (**A**) Xenopus egg extracts were either mock depleted (ΔMock) or depleted with an anti-XPol κ-specific antibody (ΔXpol κ) as described in Materials and methods. Egg supernatants after depletion were analysed by western blotting with the indicated antibodies. (**B**) Analysis of Chk1P-S344 in mock-depleted or XPol κ-depleted extracts treated with ( + ) or without ( − ) aphidicolin (15 μM) and reconstituted with 25 ng of recombinant Pol κ (R-Pol κ) performed as in **B**. Chk1 was used as the loading control. (**C**) Kinetics of DNA synthesis of mock-depleted (ΔMock; left panel) or XPol κ-depleted (ΔXPol κ; right panel) egg extracts in both the presence ( + ) or absence ( − ) of UV irradiation (800 J/m$^2$) and absence ( − ) or presence of 5 mM caffeine. The extent of DNA synthesis of both extracts is expressed as percent (%) of the input DNA added to egg extracts at time = 0. Source data for this figure is available on the online supplementary information page.

suggesting repression of the replication checkpoint. Chk1 phosphorylation after fork stalling is thought to suppress origin initiation largely by inhibiting the activation of new replication factories, thus reducing the number of active factories (Maya-Mendoza et al, 2007). Hence, these results may suggest that Pol κ- immunodepleted extracts are not able to regulate the suppression of origin firing after UV-induced fork stalling and that Pol κ is epistatic to ATR/Chk1.

Collectively, the results presented in Figures 1 and 2 show that Pol κ is required to promote Chk1 phosphorylation in response to replication fork stalling and suggest that Pol κ catalytic activity may be implicated in this process.

### Pol κ is implicated in formation of small nascent DNA in response to replication stress in Xenopus egg extracts

Next we characterized the defect in Chk1 phosphorylation observed upon removal of Xpol κ from egg extracts at the molecular level, in order to determine at which step of the checkpoint signalling pathway Xpol κ is required. We first investigated whether depletion of XPol κ resulted in removal of the main checkpoint sensors from egg extracts. As can be seen in Figure 3A, neither ATR, TopBP1, RPA, the Rad9 subunit of the 9-1-1 complex nor Chk1 was co-depleted by the XPol κ antibody, further supporting that inhibition of Chk1 phosphorylation is not due to quantitative removal of these essential checkpoint factors, similarly to what observed in human cells (Supplementary Figure S1E). DNA replication inhibitors, such as aphidicolin, cause functional uncoupling of the MCM helicase and replicative DNA polymerase activities at replication forks, resulting in the formation of long stretches of RPA-bound ssDNA. We therefore verified whether replication fork uncoupling occurs in the absence of Xpol κ by monitoring the recruitment of RPA to chromatin after replicative stress. We observed that RPA recruitment was unaffected after aphidicolin treatment in the absence

of XPol κ (Figure 3B) or upon stalling of replication forks by UV irradiation (Supplementary Figure S3C), suggesting that replication fork unwinding was not perturbed.

Recent work has demonstrated that short DNA products accumulate on ssDNA templates in response to fork stalling by aphidicolin and strongly contribute to checkpoint activation (Van et al, 2010). These DNA products are longer than the size normally synthesized by Pol α, and a subset of them are generated by Pol δ, most probably on the lagging strand. The leading strand replicative polymerase Pol ε, in contrast, does not appear to play a significant role in the synthesis of these DNA products (Van et al, 2010). Since chromosomal DNA synthesis is not inhibited upon depletion of XPol κ from Xenopus egg extracts (Figure 2D), we then determine whether Pol κ is involved in the generation of small nascent DNAs at arrested forks by analysing replication intermediates formed in Pol κ-depleted extracts upon stalling forks with aphidicolin as previously described (Van et al, 2010). To this end, we monitored the incorporation of a radiolabelled nucleotide precursor into low molecular weight DNA intermediates in both control (ΔMock) and Pol κ-depleted extracts (ΔXPolκ in the absence and presence of aphidicolin 40, 60 or 90 min after chromatin template addition (Figure 3C; Supplementary Figure S3D). Synthesis of 25–150 nt-long nascent DNAs in response to aphidicolin treatment could be observed in control extracts (ΔMock, Figure 3C; Supplementary Figure S3D), confirming previous data (Van et al, 2010). Strikingly, these short DNA products were greatly reduced in Pol κ-depleted extracts (ΔXPolκ, Figure 3C; Supplementary Figure S3D), suggesting that replication intermediates formed in aphidicolin-treated egg extracts depend upon XPol κ. This finding is in agreement with the observation that, in contrast to the three replicative DNA polymerases Pol δ, Pol ε and Pol α, Pol κ is not inhibited by aphidicolin (Gerlach et al, 2001). Importantly, we verified that this phenotype was not due to

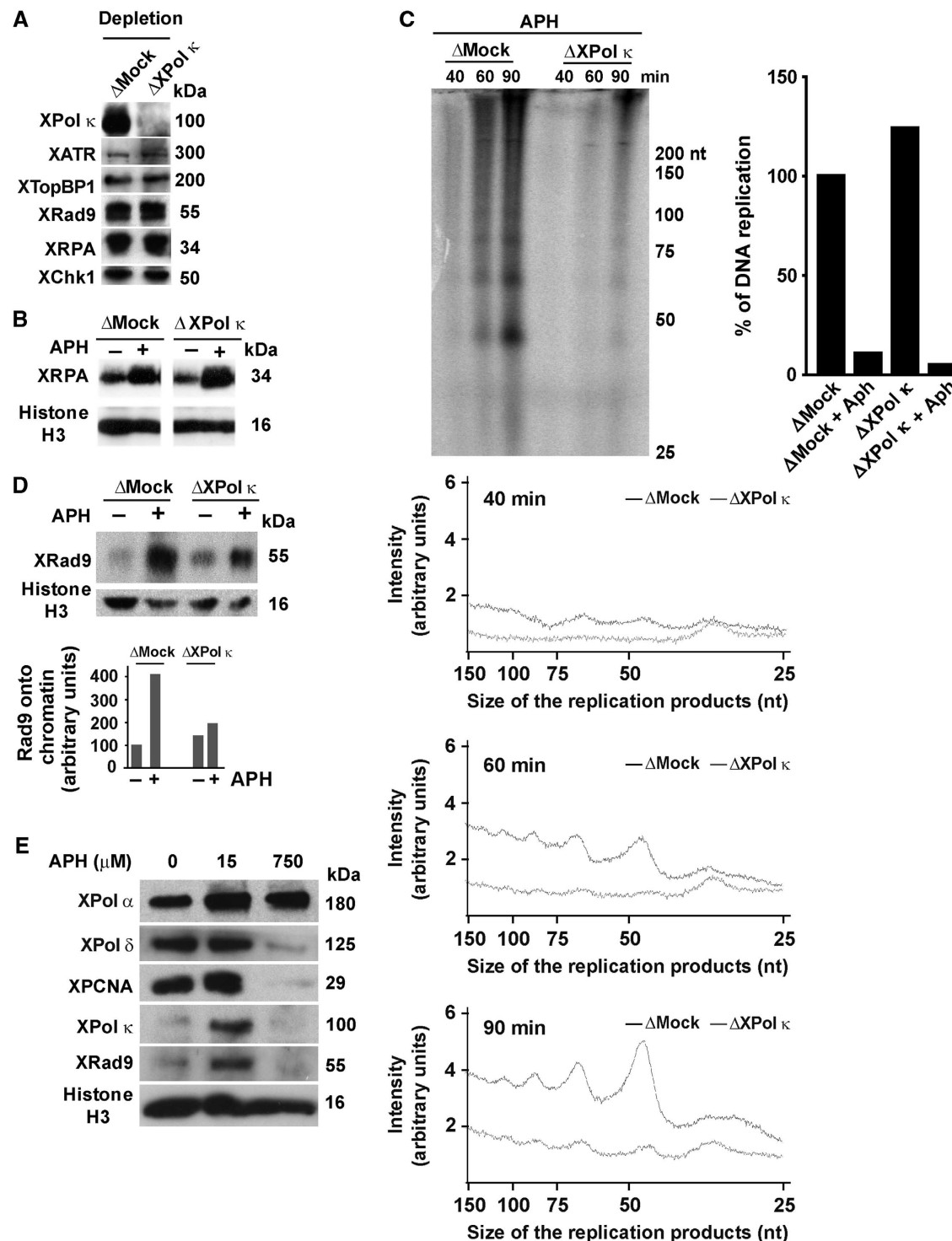

**Figure 3** Pol κ is required for the accumulation of small nascent DNA in response to replication stress in Xenopus egg extracts. (**A**) The abundance of checkpoint factors in extracts Mock-depleted (ΔMock) or XPol κ-depleted (ΔXPol κ) was analysed by immunoblotting. (**B**) Chromatin binding of RPA in egg extracts mock depleted or depleted with XPol κ antibodies in the absence (−) or presence (+) of aphidicolin. (**C**) Sperm chromatin was replicated in egg extracts mock depleted (ΔMock) or depleted with XPol κ antibodies (ΔXPol κ) containing 15 μM aphidicolin (APH) and α-[P$^{32}$]dCTP. At 40, 60 and 90 min, total DNA was purified as described (Van *et al*, 2010), replication intermediates were fractionated by denaturing polyacrylamide gel electrophoresis and detected by autoradiography after exposure to a PhosphoImager screen (Molecular Dynamics). Abundance of 25–150 nt long DNA intermediates was quantified by densitometric scanning and analysed with ImageJ software. (**D**) Sperm chromatin was incubated in control (ΔMock) or XPol κ (ΔXPol κ)-depleted egg extracts in the presence of aphidicolin (15 μM) for 60 min. Chromatin fractions were analysed by immunoblotting with the indicated antibodies. Western blot signals were quantified with ImageJ software, normalized to histone H3 signals. (**E**) Analysis of XPol κ chromatin binding. Demembranated sperm nuclei (2000 nuclei/μl of egg extract) were incubated in egg extracts for 90 min in the presence (+) or absence (−) of 15 or 750 μM aphidicolin. Chromatin fractions were obtained as described in Materials and methods and analysed by immunoblotting with the indicated antibodies. Source data for this figure is available on the online supplementary information page.

removal of Pol δ and Pol α from Pol κ-depleted extracts (Supplementary Figure S3E).

The presence of stable 5′ ends of primer-template junctions constitutes the likely binding sites for the 9-1-1 complex (Ellison and Stillman, 2003). Thus, we reasoned that the defect in synthesis of these replication intermediates in the absence of XPol κ might also affect the loading of the 9-1-1 complex onto chromatin. Therefore, we determined whether Pol κ depletion affects recruitment of the Rad9 subunit of the 9-1-1 complex to chromatin after aphidicolin treatment. We found that recruitment of the Rad9 subunit of the 9-1-1 complex after aphidicolin treatment was strongly reduced in the absence of Pol κ (Figure 3D) and that addition of purified recombinant Pol κ to egg extracts depleted of XPol κ restored recruitment of Rad9 (Supplementary Figure S3F). We next investigated the binding of Pol α, Pol δ, Pol κ, PCNA and Rad9 onto chromatin during DNA replication in the absence of aphidicolin, or in the presence of two concentrations of aphidicolin, a low dose (15 μM) that does not inhibit Pol α and a high dose (750 μM) that prevents the primer synthesis by Pol α (Byun *et al*, 2005) (see Figure 3E). The data clearly show that in the absence of aphidicolin, only the replication factors Pol α, Pol δ, and PCNA are loaded, as expected. Similarly to what observed in mammalian cells (Supplementary Figure S2A), in Xenopus egg extracts Pol κ is recruited to chromatin upon replication stress (15 μM aphidicolin) together with Rad9, further supporting the requirement of Pol κ for the replication checkpoint upon replication stress (Figure 3E). In contrast, in the presence of high doses of aphidicolin only Pol α was chromatin bound while PCNA, Pol δ and Rad 9 were not (Figure 3E), as expected (Michael *et al*, 2000; Byun *et al*, 2005; Maiorano *et al*, 2005). In these conditions, Pol κ was not chromatin bound, suggesting that similar to PCNA and Rad9, its recruitment is dependent on the primer synthesis by Pol α upon replication stress. Finally, we performed a series of immunoprecipitation experiments using Pol δ, Pol κ or Rad9 antibodies (see Supplementary Figure S3G and H). The data obtained clearly indicate that XPol κ interacts with XRad 9, but not with Pol δ nor with Pol α. Collectively, these results show that Pol κ is implicated in the synthesis of short DNA intermediates, which in turn may facilitate recruitment of the 9-1-1 complex at stalled forks and consequently contribute to efficient activation of the replication checkpoint.

### Pol κ suppresses the formation of ssDNA induced by replicative stress

Next, we monitored formation of ssDNA in S phase in human cells after HU treatment and downregulation of Pol κ. For this purpose, ssDNA was detected by immunofluorescence in PCNA-positive nuclei using a bromodeoxyuridine-based assay. In this assay only BrdU contained in ssDNA is detected by anti-BrdU antibody staining under non-denaturing conditions (Raderschall *et al*, 1999). As seen in Figure 4A and B, around 70–80% of both control and Pol κ-depleted S-phase cells (PCNA positives) are ssDNA positive after treatment with 2 mM HU for 3 h (R0). This observation indicates that in mammalian cells Pol κ depletion does not affect replication fork uncoupling, consistent with what observed in Xenopus egg extracts (Figure 3B). In addition, after HU treatment, cells were released into fresh medium and the proportion of ssDNA-positive cells 3 h after release was determined (R3 in Figure 4A and B). Surprisingly, we found that

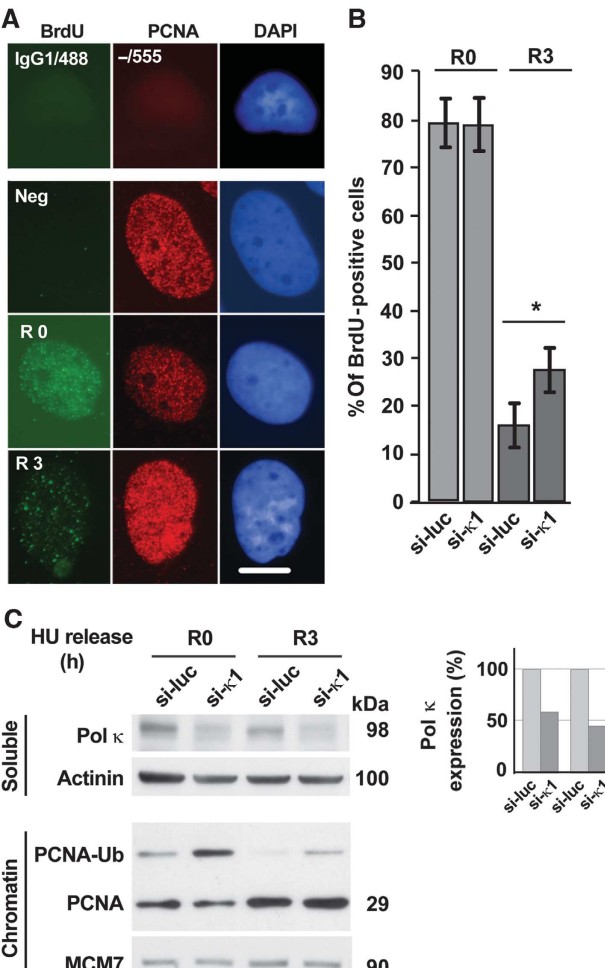

**Figure 4** Pol κ suppresses formation of single-stranded DNA induced by replicative stress upon recovery from an HU block. (**A**, **B**) Persistence of BrdU-positive cells in Pol κ-deficient cells 3 h after release from HU. Exponentially growing MRC5 cells were transfected with the indicated si-RNA; 48 h later cells were treated with 2 mM HU for 3 h and collected (R0) or released for 3 h in fresh medium (R3). Detection of BrdU (green) and PCNA (red) was performed in non-denaturing conditions. DNA was counterstained with DAPI. IgG1/488 and -/555 were controls of immunodetections. Examples of negative and positive cells for BrdU labelling at R0 and R3 are shown in **A**; scale bar 10 μm. (**B**) At least 150 MRC5 were randomly acquired with wide field microscopy and BrdU-positive nuclei in PCNA-positive nuclei were quantified in three independent experiments. Standard deviations were indicated by error bars and a *t*-test was applied (*$P<0.05$). (**C**) Increased level of Ub-PCNA onto chromatin after HU in Pol κ-deficient cells. Extracts of MRC5 cells transfected with si-luc or si-κ1 were fractionated. Soluble and chromatin fractions were then subjected to immunoblotting with the indicated antibodies. Actinin and MCM7 served as loading controls for soluble and chromatin fractions, respectively. Source data for this figure is available on the online supplementary information page.

Pol κ-depleted cells had a significantly higher proportion of ssDNA-positive nuclei compared to control cells (Figure 4B), suggesting that Pol κ is important to suppress ssDNA during the recovery from HU-mediated arrest.

To confirm this observation we used a second, independent assay that consists in monitoring the monoubiquitylated form of PCNA (Ub-PCNA). This post-translational modification is generated in response to replication stress by the ubiquitin E2–E3 complex composed of Rad6 and Rad18 and depends

upon replication fork uncoupling (Chang *et al*, 2006; Davies *et al*, 2008). As shown in Figure 4C, higher amounts of Ub-PCNA bound to chromatin were found in Pol κ-depleted cells compared to control cells, that persisted after recovery from an HU block. Collectively, these results suggest that Pol κ is implicated in suppression of ssDNA after recovery from replicative stress in human cells. Given that Pol κ activity promotes efficient phosphorylation of Chk1 after replicative stress and that DNA synthesis on the unwound template is required for checkpoint activation (Van *et al*, 2010), it is likely that these two functional events may be mechanistically linked (see Discussion).

### Pol κ depletion in unstressed cells perturbs replication dynamics and affects genome stability

It has been reported that mice defective for the *POL K* gene manifest a spontaneous genetic instability phenotype (Burr *et al*, 2006; Stancel *et al*, 2009) but the mechanisms involved in this phenotype remain thus far totally unknown. The novel checkpoint function of Pol κ might play a role in regulating genomic stability under normal growth conditions. Indeed, the replication checkpoint detects a variety of endogenous replication-impeding events and responds by triggering the ATR-mediated phosphorylation of Chk1 protein kinase, suppressing replication-associated genomic instability (Petermann and Caldecott, 2006). We addressed this issue by monitoring replication protein A (RPA) focus formation in unstressed cells, which is indicative of endogenous ssDNA accumulation due to fork stalling and/or DNA damage. Pol κ-deficient cells showed high levels of RPA focus formation (Figure 5A, left panel) and a significant increase in the number of RPA foci per nucleus (right panel) as compared with control cells containing Pol κ. Chromatin fractionation followed by immunoblotting confirmed recruitment of RPA to chromatin in otherwise unperturbed Pol κ-depleted cells by using two independent siRNAs (Figure 5B) at levels comparable to what observed after UV treatment, which served here as a positive control for replicative stress (lane UV). To verify that the additional recruitment of RPA is due to ssDNA accumulation at stalled forks and not to firing of new replication origins, we checked the recruitment of initiation factors such as Cdc45 and Pol α and observed that they remained unchanged following Pol κ depletion (Supplementary Figure S4A). In order to determine whether RPA recruitment depends upon Pol κ catalytic activity, we used a 3′UTR siRNA to deplete endogenous Pol κ, and then complemented Pol κ-depleted cells with either wild-type (catalytically active) or Dead (catalytically inactive) Pol κ. Complementing Pol κ-depleted cells with wild-type, and not with catalytically inactive Pol κ suppressed the high level of RPA recruitment to chromatin observed in Pol κ-depleted cells (Figure 5C). Expression of Dead Pol κ but not wild-type Pol κ, in the presence of endogenous Pol κ, also promoted high levels of RPA loading onto chromatin similar to what seen in Pol κ-depleted cells (Figure 5C). This result suggests that the catalytically inactive Pol κ might act in a dominant-negative fashion in the presence of native Pol κ.

In order to determine the status of the checkpoint in unperturbed cells in the absence of Pol κ, we monitored Chk1 phosphorylation in Pol κ-depleted cells. We found that Chk1 phosphorylation was not significantly enhanced in the absence of Pol κ despite accumulation of RPA onto chromatin (Figure 5B), suggesting that at least some of the spontaneous stalled replication forks cannot trigger replication checkpoint activation. These results collectively show that absence of Pol κ affects Chk1 phosphorylation even in unstressed cells and leads to RPA focus formation, suggesting the presence of stalled, damaged forks and/or unwound DNA. We observed a significant increase in γ-H2AX focus formation in Pol κ-deficient cells relative to control cells, most noticeably in S-phase cells (Supplementary Figure S4B). The increased γ-H2AX focus formation following Pol κ depletion was still observed in an NER-defective HeLa cell line (XPA$^{KD}$ cells) (Biard *et al*, 2005; Supplementary Figure S4C), indicating that phosphorylation of H2AX does not depend critically upon the NER function of Pol κ. Enhanced γ-H2AX focus formation was still observed in low (5%) oxygen conditions (Supplementary Figure S4C), arguing that cellular oxidative stress was unlikely to be the predominant stimulus for focus formation.

Pol κ depletion also affected replication factory dynamics over the duration of S phase. Pol κ-depleted cells contained predominantly middle and late phase replication factories, with fewer cells displaying early replication factory morphology compared to control cells (Figure 5D). Again, we checked that the change in replication factory dynamics was independent of endogenous oxidative stress (Figure 5D).

Together, these observations suggest that Pol κ may function at stalled forks also in unstressed cells. We thus speculated that deficiency of Pol κ in unstressed human cells would result not only in under-replicated DNA or unresolved replication intermediates in S phase, but also their persistence in mitosis in a checkpoint-blind manner. A recent study shows that under-replicated regions persisting into mitosis can be transmitted to daughter cells in 53BP1-shielded nuclear bodies (Lukas *et al*, 2011). Therefore, we analysed 53BP1 nuclear body formation in G1 (Cyclin A negative) nuclei from Pol κ-depleted cells, and we compared it to mock-depleted cells (Figure 6). We found a significant increase in the number of spontaneous 53BP1 nuclear bodies in G1 in the absence of Pol κ (Figure 6 A–C; Supplementary Figure S5B), as hallmark of incomplete DNA replication during the previous cell cycle. These observations may explain why the FACS profile of Pol κ-depleted cells displays a mild elevated fraction of cells in G2 and M phases (Figure 7A and B) which could be slightly delayed, and a slower proliferation than control cells (Figure 7C). Interestingly, increased 53BP1 nuclear body formation in G1 was also noticed in Pol κ-depleted cells after treatment with aphidicolin (Figure 6 D–F; Supplementary Figure S5C). These experiments further support that Pol κ is required to both prevent incomplete replication intermediates and activate the replication checkpoint in the presence of endogenous or external replication stress.

## Discussion

DinB/ Pol κ-like polymerases are found in all domains of life and are among the most highly conserved of all the TLS DNA polymerases (Waters *et al*, 2009). The ubiquity of Pol κ argues that this protein may contribute to additional aspects of cell physiology in addition to its role in TLS (Ogi *et al*, 2002; Avkin *et al*, 2004) and DNA repair synthesis following the NER-mediated excision of

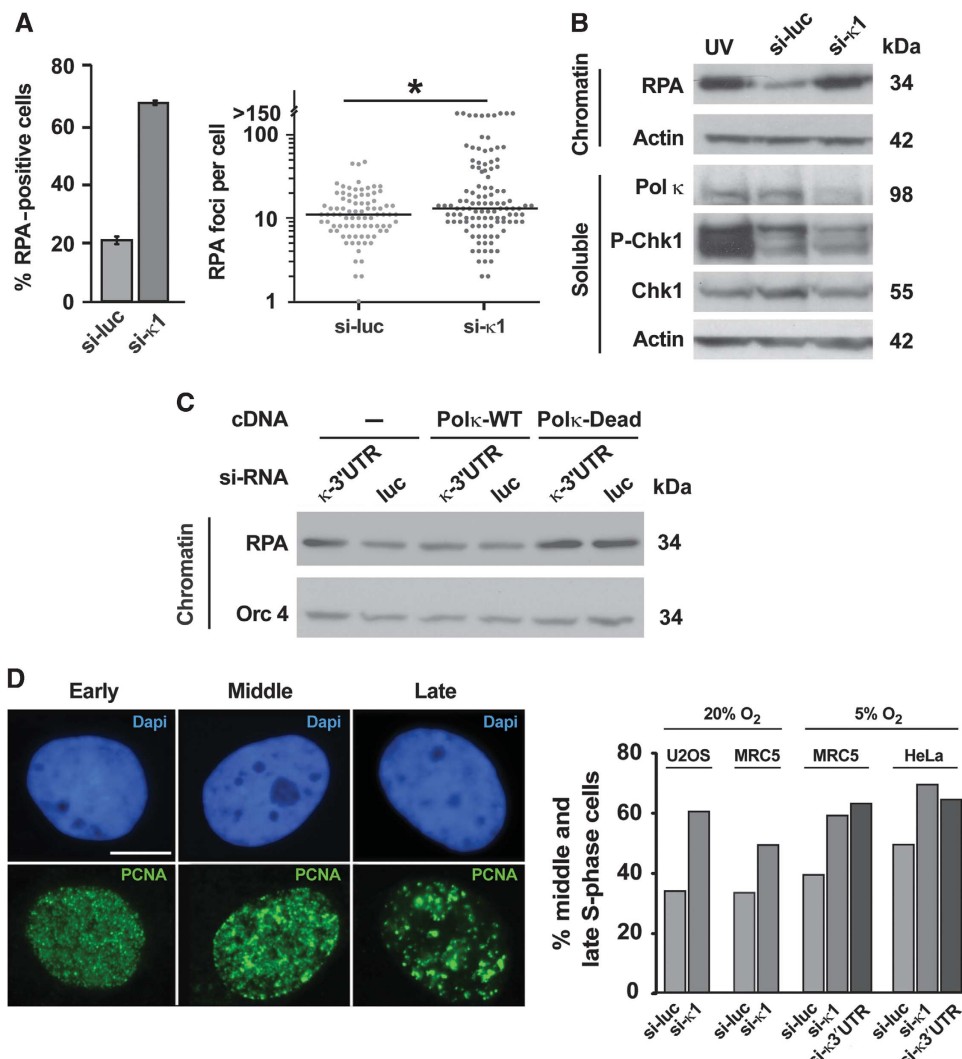

**Figure 5** Pol κ depletion triggers perturbation of DNA replication program. (**A**) Increased RPA foci formation in nuclei from Pol κ-depleted cells. Left panel: MRC5 cells transfected with control siRNA (si-luc) or siRNAs targeting Pol κ (si-κ1) were randomly acquired (*n* > 100 nuclei) by confocal microscopy and quantification of RPA-positive nuclei was performed 48 h after transfection; DNA content was visualized by DAPI staining. Right panel: cells transfected by the indicated siRNA were randomly acquired (*n* > 80 cells) with wide field microscopy and the number of RPA foci per nucleus was automatically counted with ImageJ software; the distribution of the RPA foci per cell was presented and the *P*-value determined with the non-parametric Mann–Whitney test was 0.011 (\*). (**B**) RPA accumulation onto chromatin in Pol κ-deficient cells. Extracts of MRC5 cells transfected with control siRNA (si-luc) or siRNAs targeting Pol κ (si-κ1) were fractionated. Chromatin and soluble fractions were then subjected to immunoblotting with the indicated antibodies. Extracts from the untransfected cells irradiated with UV (50 J/m$^2$) serves as positive control for RPA hyperloading and Chk1 phosphorylation; actin serves as loading control. (**C**) Catalytic activity of Pol κ is required for preventing RPA accumulation onto chromatin in unstressed cells: western blot with the specified antibodies of chromatin cell extracts prepared from HeLa cells co-transfected by the indicated siRNA and empty vector (−) or vectors expressing FLAG-tagged wild-type Pol κ (Pol κ-WT) or the FLAG-tagged catalytic inactive mutant of Pol κ (Pol κ-Dead). (**D**) Depletion of Pol κ affects the spatial organization of active replication foci. Left panel: images of early, middle and late replication PCNA foci in U2OS cells analysed by immunofluorescence; scale bar: 10 μm. Right panel: quantification of MRC5 and HeLa cells with middle/late replication foci was performed (*n* > 100 cells) for control (si-luc) and Pol κ-deficient cells (si-κ1, si-κ3′UTR) grown under 20 and 5% of oxygen. Source data for this figure is available on the online supplementary information page.

exogenous DNA damage NER (Ogi and Lehmann, 2006; Ogi *et al*, 2010). In this work, we have described a novel and unexpected role for Pol κ in response to replication stress using two different experimental systems xenopus and mammalian cultured cells. We have provided evidence that Pol κ is required for checkpoint activation after replication fork stalling with DNA polymerase inhibitors, such as hydroxyurea or aphidicolin, or in the presence of UV-blocking lesions. These effects appear to be specific to Pol κ since (i) they were observed in Pol κ−/− MEFs, (ii) depletion of Pol κ did not significantly affect expression

of other TLS or replicative polymerases, (iii) cells depleted for Pol η, one of the closest relatives of Pol κ, retained a functional replication checkpoint, and (iv) the checkpoint defect could be efficiently rescued by expression of wild-type Pol κ- in mammalian cells and by addition of a recombinant form of Pol κ in Xenopus egg extracts depleted for XPol κ.

Our findings differ from a previously published report, which showed that BPDE-induced phosphorylation of Chk1 is increased and persistent in Pol κ-deficient MEF cells (Bi *et al*, 2005). The MEF we used here (Schenten *et al*, 2002) shows much higher UV sensitivity than the MEF used by Bi

*et al* (2005). Furthermore, we observed in our Pol κ −/− MEFs a significant decreased level of Chk1 (Supplementary Figure S1F), which was not observed in the MEFs from Bi and colleagues. This strongly suggest that the discrepancy may be due to differences in MEF cell types and further studies will be required to clarify the issue.

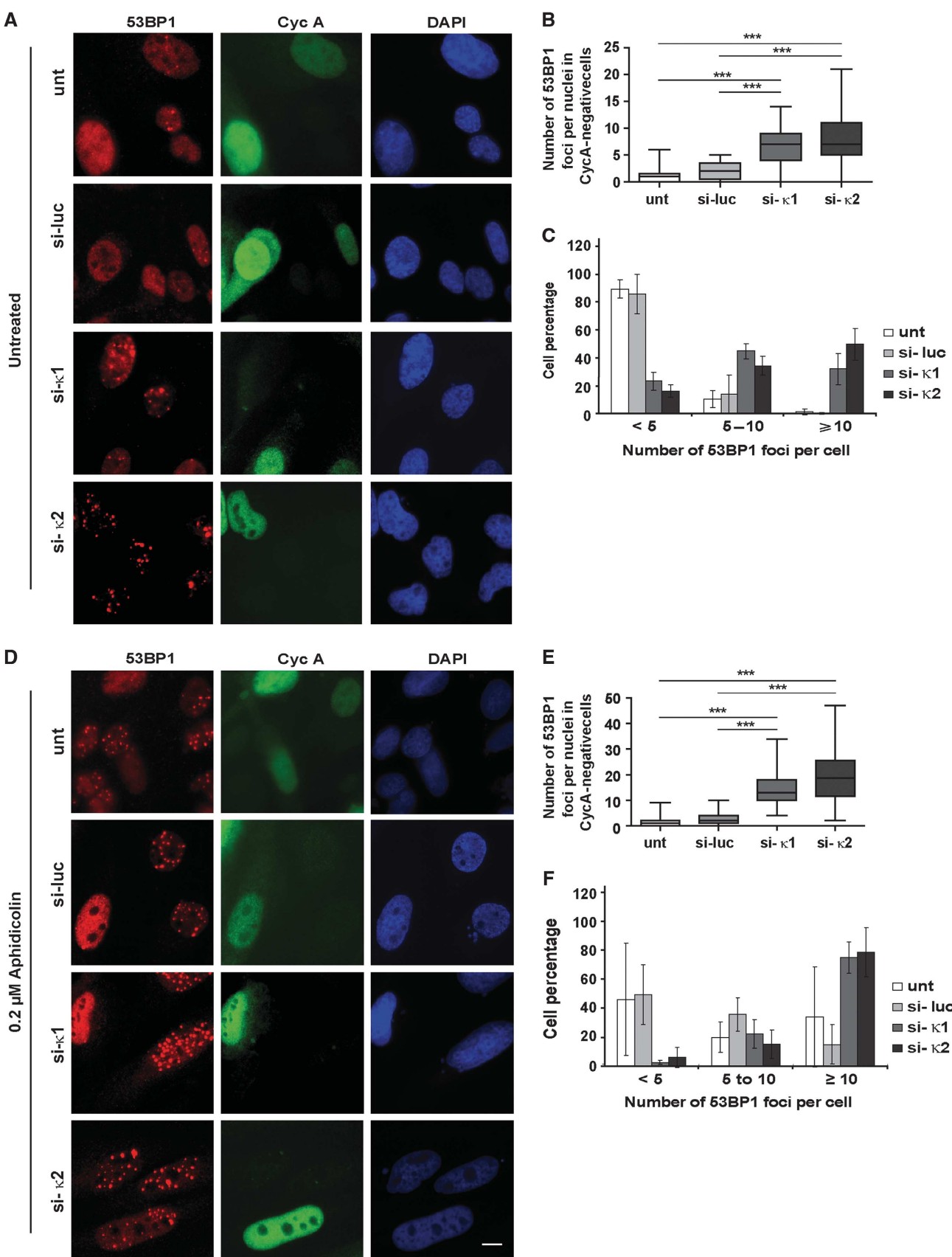

The decreased level of Chk1 that we observed in our Pol κ−/− MEFs may be explained by the chronic replication stress induced by loss of Pol κ. Indeed, the persistent absence

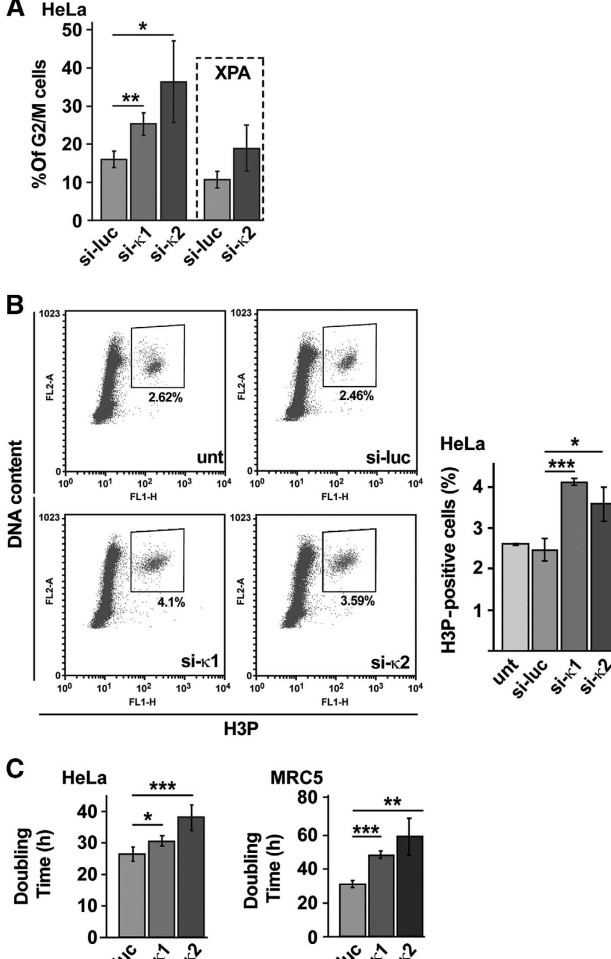

**Figure 7** Analysis of cell-cycle progression and cell proliferation in Pol κ-deficient cells. (**A**) The percentage (%) of G2/M cells was determined in HeLa or in HeLa XPA[KD] cells by flow cytometry analysis on control (si-luc) and Pol κ-deficient cells (si-κ1 and si-κ2). Experiments were done three times and standard deviations are indicated by error bars. *P*-values were calculated using Student's *t*-test (\**P*<0.05; \*\**P*<0.01). (**B**) Evaluation of the function of the G2/M checkpoint: the relative mitotic index in Pol κ-deficient cells compared with control cells was measured by monitoring the phosphorylated form of Histone H3 by flow cytometry analysis; DNA content was determined after DNA labelling with propidium iodide (PI). The percentage of H3P-positive cells was determined in three independent experiments and standard deviations were indicated by error bars. *P*-values were calculated using Student's *t*-test (\**P*<0.05; \*\*\**P*<0.001). (**C**) Doubling time calculated from growth curve done during 96 h on control and Pol κ-deficient HeLa and MRC5 cells. Experiments were done three times in triplicate and standard deviations are indicated by error bars. *P*-values were calculated using Student's *t*-test (\**P*<0.05; \*\**P*<0.01; \*\*\**P*<0.001).

of Pol κ induces a mild but chronic replication stress, triggering prolonged fork stalling that can collapse and generate DSB, which in turn can activate slightly but chronically the DNA damage checkpoint, including the phosphorylation of Chk1 which is here independent of Pol κ. Since Chk1 activation by phosphorylation induces its degradation by the proteasome degradation system (Zhang *et al*, 2009), this can explain why Chk1 levels are reduced. Interestingly, it has been recently reported that a chronic replication stress by oncogenes triggers a fast decrease in Chk1 level (Neelsen *et al*, 2013).

Activation of the replication checkpoint strongly depends on the formation of primed ssDNA at stalled forks. Upon replication stress, leading and lagging strand replication can become disengaged at an unwound fork. On the lagging strand, primers accumulate through recycling of Pol α and are elongated by Pol δ (Van *et al*, 2010). This process generates several 5′ ends primer-template junctions, as it is the case for normal fork progression, facilitating the binding of the 9-1-1 complex. The molecular determinants of checkpoint activation on the leading strand must be different since this strand is normally replicated in a highly processive manner without interruption during replication. In contrast to Pol δ, Pol ε seems to be involved at a lesser extent in the primer elongation process at an unwound fork after replication stress (Van *et al*, 2010), in agreement with the need of a distributive synthesis. Here, we have uncovered that Pol κ contributes to formation of replication intermediates at forks stalled with aphidicolin. Our results suggest that Pol κ may be directly involved in the elongation of these short DNA products on the leading strand and may be in competition with processive replicative DNA polymerases in this process. A simple model that can be proposed from our results is the following (see Figure 8): upon induction of replication stress DNA primase-Pol α is recruited to the leading strand to synthesize primers which are elongated by Pol κ in a distributive, rather than a processive manner by a replicative polymerase. It cannot be excluded that Pol κ may also function on the lagging strand in competition with DNA Pol δ. This regulation may ensure generation of multiple 5′ ended primer-template junctions to enable efficient replication checkpoint activation and is in line with the observation that DNA synthesis at arrested forks continues at a slow rate, leading to the formation of small replication intermediates that contribute strongly to checkpoint activation (Van *et al*, 2010). We have presented also evidence that in the absence of Pol κ the DNA binding of the Rad9 subunit of the 9-1-1 complex is affected after replication stress. These observations suggest that Pol κ-dependent DNA synthesis is required for Rad9 binding. This interpretation is consistent with the observation that in the absence of Pol κ short replication intermediates are poorly made, resulting in much less 5′ ends and consequently reduced recruitment of the 9-1-1 complex. Generation of multiple short DNA

**Figure 6** Increased 53BP1 nuclear bodies in G1 following Pol κ depletion. Untransfected (unt) or transfected with control siRNA (si-luc) or siRNAs targeting Pol κ (si-κ1, si-κ2) were untreated (**A**–**C**) or treated with 0.2 μM of aphidicolin for 24 h (**D**–**F**). 53PB1 was detected by immunofluorescence in G1 nuclei (cyclin A negative). Examples of positive nuclei for 53BP1 (red), Cyclin A (green) and DAPI (blue) staining are shown in **A** and **D**; **B** and **E** are representative experiments showing the distribution of the number of 53BP1 foci per cell in control and Pol κ-deficient cells (*n* = 100); Box, 25–75 percentile range; whiskers, minimum and maximum values. Mann–Whitney test was applied to compare Pol κ-deficient cells data set with the control cells (\*\*\**P*<0.001). In (**C**) and (**F**), 53BP1-positive nuclei were scored and classified in the indicated categories based on the number of 53BP1 nuclear bodies. Experiments were done three times (*n* = 100) and standard deviations are indicated by error bars.

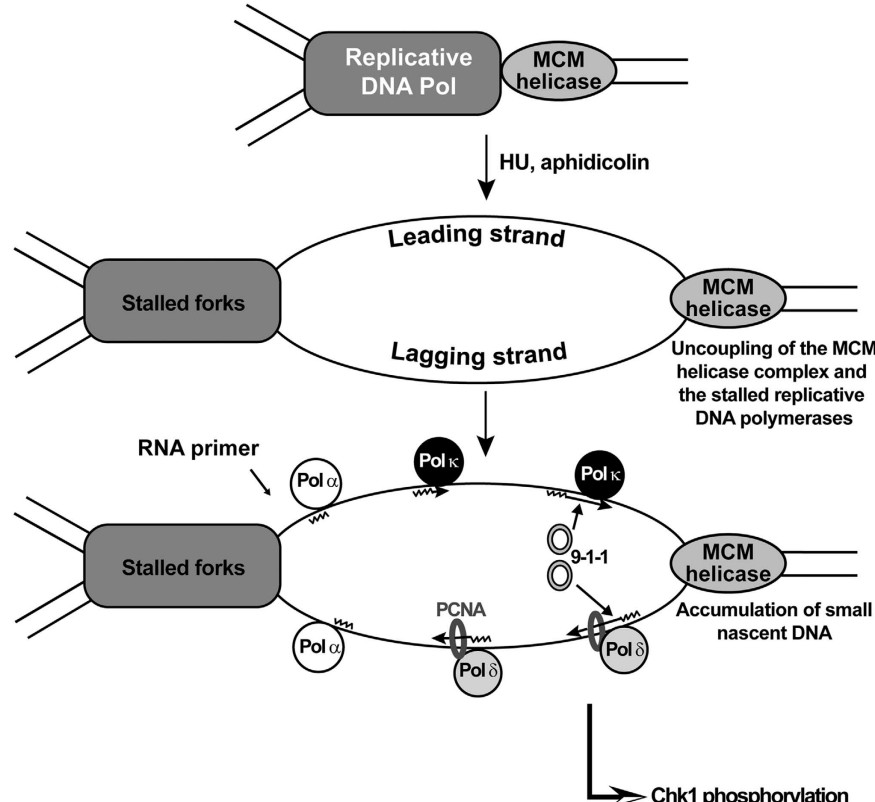

**Figure 8** A model for the checkpoint function of Pol κ. Upon replication stress, new short RNA-DNA primers continue to be synthesized and elongated on the unwound ssDNA downstream of a stalled replisome. On the lagging strand, primers accumulate through recycling of Pol α and are elongated by Pol δ with PCNA. On the leading strand, Pol α could be also recruited and elongation of the primers could be made by the distributive Pol κ in order to allow high levels of 5' ends primer-template junctions. Accumulation of short nascent DNAs on both strands results in multiple 5' ends of primer-template junctions facilitating the binding of the 9-1-1 complex and subsequently checkpoint activation.

intermediates by Pol κ may also ensure that replication restart occurs as we have observed that removal of Pol κ from human cells leads to persistence of ssDNA after recovery from a block with HU.

It is well known that Pol κ-deleted cells are sensitive to UV radiation, although this polymerase does not support replication bypass past thymine-thymine (T<>T) dimers or [6,4] pyrimidine-pyrimidone photoproducts (Okada *et al*, 2002; Schenten *et al*, 2002). These apparently contradictory results are explained by the collaboration of Pol κ with the replicative DNA polymerases Pol δ and Pol ε during the NER repair synthesis after the excision of UV damage (Ogi *et al*, 2010). Our observation that depletion of Pol κ strongly affects Chk1 phosphorylation after UV irradiation (see Supplementary Figure S2B) offers additional insight into the role of Pol κ in the response to UV damage.

Pol κ has been previously considered to be involved in the maintenance of genomic stability in the absence of external stress. Mice defective for the *POL K* gene manifest spontaneous genetic instabilities including elevated frameshift mutations in the germline at tandem repeat minisatellite loci (Burr *et al*, 2006; Stancel *et al*, 2009). Since the replication checkpoint is known to suppress also genomic instability by detecting and responding to a variety of endogenous replication-impeding events (Petermann and Caldecott, 2006), minisatellite instability observed following Pol κ depletion is consistent with prolonged replication fork pausing at these natural replication barriers, due to

defective replication checkpoint signalling, delayed checkpoint recovery and replication restart with resulting increased mutagenesis. These findings prompted us to examine whether the novel checkpoint function of Pol κ might play a role in regulating genomic stability under normal growth conditions. We found that in unperturbed conditions Pol κ-depleted cells have proliferative defects, altered replication factory dynamics, spontaneous replication stress, and under-replicated regions which are transmitted to daughter cells. Altogether, these observations reveal a previously unrecognized role for Pol κ during DNA replication in unperturbed cells, which appears to depend upon the catalytic activity of Pol κ, as demonstrated by a combination of depletion and complementation experiments, and support its checkpoint function in the absence of external stress, extending the role of this DNA polymerases outside the TLS. Pol κ is believed to be an error-prone enzyme, which may generate mutations when it acts on undamaged templates (Ohashi *et al*, 2000). Pol κ may however have higher fidelity *in vivo*, as suggested by recent work showing that Pol κ can perform accurate microsatellite DNA synthesis (Hile *et al*, 2012).

In conclusion, the concept that TLS could be not the sole function assigned to the Y-family DNA polymerases is now emerging as we recently demonstrated that Pol η is required for the stability of common fragile sites during unperturbed S phase (Rey *et al*, 2009; Bergoglio *et al*, 2013). Our work provides new mechanistic insights into the roles of Pol κ in

responding to DNA replication fork stalling by endogenous barriers or after external stress to ensure high cell viability and genomic stability.

## Materials and methods

### Cell culture, cell lines, proteins and plasmids
HeLa, MRC5, and U2OS cells were cultured as described previously (Betous *et al*, 2009; Rey *et al*, 2009). 293T cells were cultured in DMEM/Glutamax with 10% SVF (Lonza) and 1X antibiotics (GIBCO). When indicated, cells were cultured at 37°C in a humidified incubator in an atmosphere containing 5% $O_2$ (Sanyo incubator). The human Pol κ coding sequence was cloned into the pcDNA3.1Flag vector (Invitrogen). The catalytically inactive mutant of human Pol κ (Dead Pol κ) was constructed by using Quick Change (Stratagene) according to manufacturer's instructions to incorporate mutations (D198A and E199A). When indicated, cells were treated with hydroxyurea, Bromodeoxyuridine (both from Sigma) or UV irradiation (UVC light meter, Digit Instrument) for doses and time indicated in the figure legends. Purified full-length recombinant Pol κ (RPol κ) (98 kDa) used in rescue experiments in Xenopus egg extracts was purchased from Enzymax (Lexington, KY).

### Xenopus methods

*Egg extracts preparation, immunodepletion, immunoprecipitation and replication assay.* Xenopus sperm chromatin and egg extracts (LSS) were prepared as previously described (Murray, 1991). For immunodepletion experiments, XPol κ antibodies were covalently coupled to recombinant Protein A beads (GE Healthcare) and complete depletion of Pol κ was achieved by incubating one volume of egg extract with 40% of antibodies (Vol/Vol) for 40 min twice at 4°C on a rotating wheel. Sperm chromatin (2000 nuclei/μl of egg extract) was added to egg extracts and incubated at room temperature for the indicated time. For immunoprecipitation, egg extracts were supplemented with cycloheximide and diluted 10-fold in ice-cold Xb buffer in the presence of proteases inhibitors (see next paragraph). Antibodies (10 μg each) were added and incubated at 4°C on a rotating wheel for 1 h. Then, 20 μl of Protein A sepharose beads was added and incubation was prolonged for 1 h. ProteinA beads-immunocomplexes were collected by low speed centrifugation at 4°C and washed several times in Xb as above. Bound proteins were eluted with Laemmli buffer. For replication assays, radiolabelled DNA samples were prepared by addition of α-[$^{32}$P]dCTP (3000 mCi/mmole, Perkin Elmer) to the extract supplemented with demembranated sperm chromatin, cycloheximide (250 μg/ml) and an energy regeneration system (1 mM ATP, 2 mM $MgCl_2$, 10 mM creatine kinase, and 10 mM creatine phosphate). Reactions were performed with or without aphidicolin or caffeine (both from Sigma), stopped by addition of stop buffer (0.5% SDS, 20 mM EDTA, pH 8.0, and 500 μg/ml proteinase K) and analysed by denaturing polyacrylamide gel electrophoresis as described (Van *et al*, 2010).

*Chromatin isolation from egg extracts.* Egg extracts supplemented with demembranated sperm nuclei were diluted 10-fold with ice-cold Xb buffer (10 mM Hepes pH 7.7; 100 mM KCl; 50 mM sucrose; 2 mM $MgCl_2$, 5 μM leupeptine, aprotinin and pepstatin) and centrifuged at 1500 g in a Sorvall centrifuge at 4 °C for 5 min to sediment nuclei. Nuclei were washed once in ice-cold Xb and detergent extracted with 0.1% NP-40 for 5 min on ice. Chromatin (pellet) and soluble nucleosolic (supernatant) fractions were obtained by centrifugation at 6000 g for 5 min at 4°C in a microfuge.

### siRNAs, transfection, chromatin fractionation, western blotting and quantification
To knock down the *POL K* gene, cells were transfected with two independent siRNA (Betous *et al*, 2009) or Pol κ siRNA Smart Pool (Dharmacon) directed either the coding sequence (D-siκ) or the 3'UTR region (si-κ3'UTR). For control, siRNA against luciferase (Elbashir *et al*, 2001) was used. For siRNA transfection experiments, $1 - 1.5 \times 10^5$ cells were seeded 24 h before transfection with 100–300 nM siRNA using lipofectamine 2000 (Invitrogen). For vectors transfections, 2 μg of DNA was transfected with JetPrime reagent (Polyplus-Ozyme). Chromatin fractionation was performed as in Zou *et al* (2002) with the following modifications: solution A (10 mM PIPES pH 7, 300 mM sucrose, 3 mM $MgCl_2$, 150 mM NaCl, 1 mM EGTA, 1 mM DTT, 0.1%, Triton X-100, 0.5 μg/ml Pepstatine and 1X to 3X HaltTM protease/hosphatase inhibitors; Thermo Scientific) was used. Cells were left on ice during 5 min then spun 3 min at 7500 r.p.m. at 4°C. Pellets were incubated in solution A for 5 min on ice, spun again, and resuspended in solution A (Chromatin fraction). The supernatants from the two extraction steps were pooled (soluble fraction). Chromatin fractions were sonicated. Whole cell extracts were performed as previously described (Betous *et al*, 2009). Blots were detected by ECL Western Blotting Substrate (Pierce) or ECL Plus Western blot detecting system (GE Healthcare). Where indicated, western blots were quantitated using ImageJ software.

### Immunofluorescence
For BrdU, PCNA, RPA immunodetection: cells were washed in CSK buffer (10 mM PIPES pH 7.5, 100 mM NaCl, 300 mM Sucrose, 3 mM $MgCl_2$, 1X HaltTM protease/phosphatase inhibitors; Thermo Scientific) and pre-extracted in CSK-0.5% Triton X-100 for 2 min, rinsed in CSK buffer and PBS, fixed in 2% paraformaldehyde for 20 min. Coverslips were blocked in PBS containing 5% BSA and 0.1% Triton for 15 min. Antibodies were diluted in PBS containing 1% BSA and 0.1% Triton. Incubation times were 90 min for primary. Nuclei were counterstained wit 4',6'-diamino-2-phenyl-lindole (DAPI). Coverslips were mounted in Vectashield Hard Set (Vectors Laboratory). BrdU (Becton Dickinson, 1:100), PCNA (Abcam, 1:800), RPA (Calbiochem, 1:200), For γ-H2AX immunodetections: cells were fixed in 3.2% paraformaldehyde, 2% sucrose for 15 min, permeabilized for 4 min in Triton buffer (0.5% Triton X-100 in 20 mM HEPES pH 7.4, 50 mM NaCl, 3 mM $MgCl_2$, 300 mM sucrose). γ-H2AX (Millipore, 1:500) incubations were performed as previously described. For 53BP1 and Cyclin A immunodetections: cells were fixed in 4% paraformaldehyde for 20 min, permeabilized for 5 min in 0.25% Triton X-100/1X PBS, incubated for 20 min in 5% goat serum and 2 h with primary antibodies, 53BP1 (Abcam, 1:300), CyclinA (Santa Cruz, 1:150), Mouse IgG1 (Southern Biotech, 1:150) and Rabbit IgG (Cell Signaling, 1:300). Secondary antibodies were incubated 30 min at room temperature, anti-mouse Alexa-fluor 488 and anti-rabbit Alexa-fluor 555 (Molecular Probes, 1:1200).

### Microscope image acquisition
Images acquisition of multiple random fields were carried out on a wide field Leica DMLA microscope equipped with × 63 oil immersion objective (PLAPO 1.4 Leica) using a Cool-Snap HQ CCD camera (Photometrics, Roper Scientific) driven by PM capture Pro6.0 or Metamorph software. Images were assembled with Adobe photoshop and Illustrator. When indicated in the figure legends, images of RPA were acquired using the × 100 NA 1.4 objective of a SP2 confocal microscope (Leica microsytems) using laser lines 488 and 633 nm for excitation of Alexa Fluor 488 and ToPro3 dyes.

### Cell-cycle analysis and measurement by flow cytometry
In all, 100 000 cells were seeded 48 h prior to harvesting. Cells were washed in PBS once and 300 μl of Propidium Iodide solution (25 μg/ml Propidium Iodide (Sigma), 0.1% $C_6H_5O_7Na_3$, $2H_2O$, 10% RNAse A (Sigma), 0.1% Triton X-100) was added 30 min prior to FACS analysis (FACScan, Becton Dickinson). Results were analysed by using Modfit software. Analysis γ-H2AX and H3P by flow cytometry were performed as described (Betous *et al*, 2009), H3P (Cell Signaling, 1:75).

### Supplementary data
Supplementary data are available at *The EMBO Journal* Online (http://www.embojournal.org).

## Acknowledgements

We thank Renaud Poincloux and Françoise Viala (IPBS, CNRS Toulouse) for their technical contributions. We thank S Waga (Japan Women's University) for anti Xpol δ antibodies. This work was supported by La Ligue contre le cancer (Equipe Labellisée 2013 to JSH; PhD grant for RB), 2 INCa Grants (CHECKPOL-2007

and YPOL-2010 to JSH), 1 ARC grant (3156 to DM), and FRM (FRM Equipes to DM).

*Author contributions*: RB, MJP, DM and JSH designed the experiments. RB, MJP, LP, SVdL and EOS performed the experiments in human cells. BR, MJP and DM performed the experiments in Xenopus extracts. NN, PG and TN performed the production of the human Pol Kappa antibodies. RB, MJP, CG, EF, CC, DM and JSH analysed the data and contributed to the manuscript preparation. DM and JSH supervised the project and wrote the manuscript.

## Conflict of interest

The authors declare that they have no conflict of interest.

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
