## [Review Process File · The EMBO Journal]

Manuscript EMBO-2012-83985

DNA polymerase kappa - dependent DNA synthesis at stalled replication forks is required for CHK1 activation

Rémy Bétous, Marie-Jeanne Pillaire, Laura Pierini, Siem Van der Laan, Bénédicte Recolin, Emma Ohl-Séguy, Caixa Guo, Naoko Niimi, Petr Gruz, Takehiko Nohmi, Errol Friedberg, Christophe Cazaux, Domenico Maiorano, Jean-Sébastien Hoffmann

Corresponding author: Jean-Sébastien Hoffmann, Centre National de la Recherche Scientifique

Review timeline:	Submission date:	21 November 2012
	Editorial Decision:	19 December 2012
	Appeal:	10 January 2013
	Editorial Decision:	18 January 2013
	Revision received:	02 May 2013
	Editorial Decision:	22 May 2013
	Revision received:	30 May 2013
	Accepted:	04 June 2013

Editor: Hartmut Vodermaier

Transaction Report:

1st Editorial Decision

19 December 2012

Thank you for submitting your manuscript for consideration by The EMBO Journal. It has now been reviewed by three expert referees, whose reports you will find copied below. I am afraid that these reports, and the major concerns raised therein, offer insufficient support for publication of the study in The EMBO Journal, at least at the present stage. As you will see, while the reviewers acknowledge the potential interest of your observations on Pol kappa replication checkpoint roles, they remain unconvinced that your present set of data has provided conclusive evidence for this model, and sufficiently deeper understanding of this Pol kappa role also in relation to a variety of earlier findings. Of particular importance in this respect are the apparent conflicts and discrepancies with the recent reports on the involvement of other (replicative) polymerases in primer synthesis for checkpoint activation, as published by Van et al. Unless these discrepancies - which have the strong potential to undermine the key conclusions of the present study - can be decisively reconciled, we feel that the manuscript unfortunately remains not a strong candidate for an EMBO Journal paper. I therefore see little choice but to return it to you at the present stage with the conclusion that we would have to consider it premature to offer publication of a revised manuscript. I am sorry we cannot be more positive on this occasion, but nevertheless hope that you will for now find our referees' comments and suggestions helpful for your further proceedings with this study.

REFEREE REPORTS:

Referee #1 (Remarks to the Author):

In this manuscript the authors propose a role for Pol K in checkpoint activation. Unfortunately, the authors do not provide sufficient data to prove that Pol K is involved in checkpoint activation. In addition half of the results shown - fig 5, 6, 7 - do not add much to the major point of the paper. Finally, they did not put their work in the context of the literature, especially in relationship to Van et al. (Specific point 7)
For these reasons I think that the manuscript is not suitable for publication.

Specific points:

- 1) Fig1. The authors fail to provide details on how many times the experiments have been performed (especially in Figure 1c, which includes the rescue). Also, they did not monitor the level of proteins involved in checkpoint activation following siRNA treatment.
- 2) Endogenous levels of Pol K should be measured and compared to Pol alpha. It is indeed likely that the proposed role for Pol K is instead played by Pol alpha, whose concentration in egg extract is likely to be higher than Pol K
- 3) Figure 2B: A control with p-Chk1 after UV is missing.
- 4) Figures 2B/2C: the loading control must be total Chk1.
- 5) Figure 2D does not prove that depletion of Pol K does not affect DNA replication as it is claimed in the text. mock and PolK must be on the same graph with the same scale. If the scale is the same between the two graphs shown, the authors need to provide an explanation for the observation that Pol kappa depleted extract replicates ~20% better than mock. Also the graph lacks error bars.
- 6) Figure 3C. As control, the authors should provide a replication assay of the experiment to ensure that the total incorporated dCTP was similar between Mock and depleted extract.
- 7) The authors' model about the relationship between Pol delta and Pol K, regarding the generation of the short DNA fragments, which are thought to activate the checkpoint, is not supported by their results. Van et al showed that the generation of short DNA fragments is almost completely dependent on Pol delta, while the authors show that it is almost completely dependent on Pol K. This discrepancy should be experimentally addressed.
The authors should repeat the Pol delta experiment and verify the role of Pol delta in checkpoint activation in their conditions to confirm or disprove their hypothesis. This is a crucial point that needs to be addressed. Also, an epistasis analysis is required to prove that Pol delta and Pol kappa are on the same/or parallel pathways. Finally, If each polymerase is acting on a specific strand (as the authors propose), I would expect a partial reduction in the single depletion and a total reduction in the double depletion. This experiment should be performed to prove their point.
- 8) Is it possible that Pol K depletion removes some factors required for pol delta-dependent DNA synthesis? This should be discussed.
- 9) I struggle to understand the interpretation of the ssDNA experiment (Figures 4A/4B). If Pol K is required for DNA synthesis in the presence of HU, then I would expect more ssDNA in the Pol K siRNA at the R0 point. Also, if Pol K siRNA also causes a checkpoint activation defect, then I would expect even more ssDNA, given that without the checkpoint, silent origins are free to fire. The result at the R3 point shows that ssDNA persists slightly longer after removal of HU, which suggests that the absence of Pol K affects replication fork restart, rather than DNA synthesis DNA during the HU/aphidicolin arrest.
- 9) The authors claim that Pol K's function is required for checkpoint activation and that ubiquitinated PCNA is enriched on chromatin in these conditions. This is confusing given that others (Chang et al, 2006, cited in the text) demonstrated that checkpoint activation is independent of PCNA

ubiquitination.

10) Figure 5B. To ensure that the additional recruitment of RPA is due to ssDNA accumulation at stalled forks, the authors should verify that the recruitment of replication factors such as Cdc45 or Pol alpha is similar between control and Pol kappa siRNA treated cells.

Referee #2 (Remarks to the Author):

Betous and colleagues use siRNA in human cells and immunodepletion in frog egg extracts to study the role of PolK in activation of the replication checkpoint. The data show that loss of PolK compromises checkpoint activation, as measured by Chk1 phosphorylation. Follow-up experiments in frog egg extracts show that primer synthesis during a replication stress response is attenuated upon removal of PolK. Taken together, these data provide evidence for a role for PolK in checkpoint activation through primer extension, and as such they are worthy of publication once the minor issues detailed below have been taken care of.

Minor points

Introduction, 2nd paragraph. The authors state: "Independently, an alternative RFC-containing Rad17 complex recruits the heterotrimeric 9-1-1 complex to primers synthesized by DNA polymerase alpha, which in turn recruits the ATR activator TopBP1". The statement suggests a linear recruitment pathway where 9-1-1 loads first and then recruits TopBP1. Unfortunately, this statement is frequently made when summarizing checkpoint complex assembly, and there is little if any available data that support it. By contrast, two papers have shown that TopBP1 is required to recruit the 9-1-1 complex to sites of replication stress (Yan and Michael, JCB 184, p. 793, 2009 and Gong et al, Mol. Cell 37, p.438, 2010). I suggest that authors revise this sentence to reflect the currently available data.

In Figure 2D DNA replication data are presented and plotted as "% of replication" over time. What does "% of replication" mean? How can we compare the replication efficiency of the two extracts (mock- and PolK-depleted)? This should be clarified.

In Figure 3 a series of critical functional assays are performed with PolK-depleted extract, yet no control blots are shown that attest to the efficiency of depletion. Furthermore, no rescue of depleted extracts with recombinant PolK is shown for these experiments. These important controls should be included.

The blot in Figure 3D shows a ~50% reduction in the amount of Rad9 loaded onto aphidicolin-treated chromatin. Was this consistently observed? The authors should comment on this as the result has implications for leading vs lagging strand loading of 9-1-1 by PolK.

The blot in Figure 5B uses actin as a loading control for the chromatin fraction (2nd panel down). This is an odd choice for a chromatin loading control and requires explanation.

In Figure 5B the G2/M checkpoint is analyzed, but I see no indication that cells were induced to activate a checkpoint response (eg by HU or UV treatment). If these are indeed unperturbed cells then the experiment needs to be repeated with HU or UV to be meaningful.

Referee #3 (Remarks to the Author):

Betous et al

DNA polymerase kappa - dependent DNA synthesis at stalled replication forks is required for checkpoint activation

This manuscript examines the role of the translesion DNA polymerase kappa in activation of the S phase checkpoint. A potential link has been suggested by the previously documented association of pol kappa with RAD9. There has been significant recent interest in the role of primer synthesis downstream of the replication fork stalls, thus far attributed to the action of pols alpha and delta. Here the authors suggest that pol kappa may also play a role in this primer synthesis and in the subsequent recruitment of the 9-1-1 checkpoint clamp, needed for efficient initiation of the S phase checkpoint. Thus, pol kappa is required for induction of Chk1 phosphorylation following arrest of replication with hydroxyurea and for recovery from replication stress. These are interesting and potentially significant observations. However, I feel a number of problems need to be addressed further.

1. The work appears to conflict with previously published data from Vaziri's group (Bi et al JBC 2005; Bi et al MCB 2006) who showed pol kappa-deficient MEFs exhibit a protracted S phase arrest following treatment with BDPE. However, this effect was not seen with HU. While the authors address this in the Discussion, the discrepancy remains rather inadequately investigated. Why would the proposed model not apply to both HU-induced arrest and to arrest caused by defective TLS? In other words it should not really make a difference to the proposed role for pol kappa whether the lesion causing the arrest also requires pol kappa for its replication. A possible resolution to these differences is the use of pol kappa deficient MEFs by Vaziri's group. It think it important to know how such cells behaved in the authors' experiments to know whether the difference simply reflects different cell types. Experiments on pol kappa -/- MEFs, which I believe have been produced by some of the present paper's authors, would also avoid any problems of incomplete pol kappa depletion.
2. The authors propose a model in which pol kappa helps create a DNA substrate to which the 9-1-1 clamp can be recruited. How do the authors imagine pol kappa is recruited to perform this synthesis downstream of the arrested fork given that PCNA ubiquitination does not seem to be required for activation of the replication checkpoint.
3. The diminution of CHK1 phosphorylation is rather variable between cell lines and in some cases the level of CHK1 itself appears to go down to. Why is this?
4. In some blots, loading controls are missing (e.g. Fig 1D and E), while in others loading appears to be rather variable (e.g. Figs 2B and C).
5. I don't find the difference in the number of RPA foci per cell between the control and pol kappa knockdown very convincing. What does this mean when compared with the total number of RPA positive cells?
6. Some of the blots are very cropped and it would be nice to see more, especially of CHK1. Are the phosphorylated forms of CHK1 visible and do they correlate with the phosphoCHK1 signal.

Minor comments:

Abstract, line 8: 'stalled forks'

Page 4, line 2: extra '('

Page 16, line 4: extra '-' after pol kappa.

Appeal

10 January 2013

We thank you very much for having considered our manuscript by Bétous et al., "DNA polymerase Kappa is implicated in continued DNA synthesis at stalled replication forks and is required for checkpoint activation" (manuscript EMBOJ-2012-83985) for publication in EMBO Journal. We have greatly appreciated your effort for having coordinated the reviewers' reports that we have now thoroughly analyzed.

Although reviewer #1 opinion (which is most arguable throughout most of the criticisms raised, see below) is against publication, reviewers #2 and #3 find the work of interest and of importance in the field and recommend publication after taking into consideration the points they have raised, and that we are very much willing to satisfy. Reviewer #2 has only a few minor points to be clarified, some of them are just clarifications of experiments performed in *xenopus* egg extracts as it appears that he/she 's not an expert with this system. Reviewer #3 requires a clarification of previous data reported by Bi et al., using polk-/- MEFs showing activation of the checkpoint after exposure to a DNA damaging agent (BPDE) in these cells, and we would be very happy to address and explain the apparent discrepancy and include new data with the Pol κ MEFs that confirm that Pol κ is required

for the activation of the replication checkpoint after HU, as we found in human cells and *Xenopus* extracts.

As for reviewer #1, we do not understand the meaning of most of the points raised (point 2, 3, 5, 6, 9a and 9b), and the other remaining points are very arguable and not justified, some of them can be satisfied (see comments below point by point at the end of our letter).

The major point the reviewer #1 raises is the relationship between Pol κ and the replicative Pol δ regarding the generation of the short fragments at stalled replication forks after aphidicolin, important for the checkpoint activation. He/She pointed out (see point 7 of reviewer#1) the apparent discrepancy with the work of Van et al., (JCB. 2010) which showed the requirement for Pol δ in the synthesis of these small replication intermediates. In particular the reviewer argues that “Van et al. showed that the generation of short fragments is almost completely dependent on Pol δ , while the authors show that it is almost completely dependent on Pol κ ”. This argument is not justified, since depletion of neither Pol δ nor Pol κ results in a complete disappearance of the short fragments (see Fig 5C-lane Δ Pol δ /90 min in Van et al. and Fig 3C from our manuscript, lane 90 min), nor in a total decrease of P-Chk1 (see Fig 6B from Van et al and Fig 1A-B from our manuscript). Our observations show that, besides Pol δ , another polymerase is implicated at replication forks stalled after aphidicolin treatment, and underline the complexity of the system. According to us, there is no discrepancy between the two works. Moreover, as also stated in our manuscript, it has to be remembered that this finding is in agreement with the observation that, in contrast to the three replicative DNA polymerases Pol δ , Pol ϵ , and Pol α , Pol κ is not inhibited by aphidicolin, and therefore is one DNA polymerase active at arrested forks. We cannot exclude that both polymerases act in a mutually dependent and coordinative manner. We do not think that repeating the experiment described in Van et al., consisting in removing Pol δ from egg extracts, as asked by the reviewer #1, is justified and will be very informative. By the way, in Supplementary Figure 2A and Figure 3E of our manuscript we have shown that Pol κ binds to chromatin in S-phase **only** in the presence of aphidicolin, which demonstrates that loading of Pol κ occurs after that of Pol δ since Pol δ is already bound to chromatin in S-phase. Moreover, it is known that loading of the 9-1-1 checkpoint clamp requires formation of replication intermediates long enough to be recognized by the RFC-Rad17 complex that loads the 9-1-1 clamp (Ellison and Stillman 2003. Plos Biology). These intermediates are made by the concerted and sequential action of Pol α (RNA primers) and Pol δ and it is very unlikely that Pol κ can do this job independently of Pol δ since Pol κ is not processive and can synthesize only very few nucleotides. We believe that Pol κ functions on the replication intermediates made by Pol δ to restrain excessive DNA synthesis and to stabilize the short replication intermediates that are bound by the 9-1-1 complex. It is likely that in the absence of Pol κ these intermediates are not stable for reasons that we still do not understand and this model will require more work to be proven, and this is not in the scope of the paper. We can further clarify this point in the discussion.

In conclusion we do not really understand the rationale behind the reviewer #1's request. These additional data will be anyway redundant with those provided by Van et al., and will certainly diminish the originality of the paper. The aim of our work has been to demonstrate a novel role for a translesion DNA polymerase (Pol κ) in checkpoint activation, which is a completely novel concept in the field, and not to study the epistasis of this polymerase with respect to the replicative DNA polymerases. Nevertheless, we can propose a couple of experiments that would give further insights into the relationships between Pol δ and Pol κ . The first one consists of using a high concentration of aphidicolin in *xenopus* egg extracts. In these conditions only Pol α and not Pol δ , neither PCNA, nor the 9-1-1 load onto chromatin at arrested forks, as previously shown by some of us and other labs (Michael et al., 2000 Science; Maiorano et al., Cell 2005; Byun et al 2005, Genes & Dev., Zembutsu and Waga, 2006 Nucl. Acids Res.). We can determine whether in these conditions Pol κ binds or not to chromatin. The second set of experiments would be to determine whether there's a physical interaction between Pol κ and Pol δ by immunoprecipitation and whether removal of Pol κ also removes Pol δ .

In conclusion we are very much willing to provide more evidence required to convince the reviewer#1 that Pol κ is indeed an important element of the replication checkpoint and we are therefore writing to ask whether after taking into consideration all our arguments, you will be willing to give us the possibility to resubmit a revised version of our manuscript for publication in

EMBO Journal and/or to contact a fourth reviewer who would be an expert of the *Xenopus* in vitro system (such as Dr. Karlen Cimprich, Stanford University, USA, for instance).

Point by point answer to referee 1

Point 1. The experiments shown throughout the paper are representative examples of independent observations made several times and by different researchers. The level of proteins involved in the checkpoint has been checked after Pol κ si-RNA and can be showed in a revised version.

Point 2. We do not understand at all this point. Even if the concentration of the primase Pol α would be higher, we removed only Pol κ and we demonstrate in Figure 2D that Pol κ is indeed not required for replication, which rules out the unlikely possibility that Pol κ and Pol α interact. Nevertheless we can investigate this point by immunoprecipitation.

Point 3. Obviously, we have this data as previously shown by us and other lab (Recolin et al., 2012; Nucl. Acids Res.; Byun et al., 2005; Genes & Dev.; Lupardus et al., 2000 Genes & Dev.) and we can add this data to satisfy the referee.

Point 4. We can add Chk1 as a loading control.

Point 5. We do not understand this point either. Figure 2D clearly illustrates that the scale between the two graphs is identical. This experiment also shows that removal of Pol κ does not affect the rate of DNA synthesis and demonstrates that indeed Pol κ is not required for chromosomal DNA synthesis in the absence of aphidicolin, and this is also true in mammalian cells. We do reproducibly observe slightly more DNA synthesis in Pol κ -depleted extracts, which is very likely due to failure to activate the intra S phase checkpoint that restrains activation of late clusters of replication origins (Shechter et al., 2004. Nat. Cell Biol.). This is entirely consistent with the observation that caffeine, an inhibitor of ATM/ATR, has no effect on DNA synthesis after removal of Pol κ (Figure 2D).

Point 6. Figure 2D shows a representative replication assay performed with either mock- or Pol κ -depleted extracts, and also shows that indeed removal of Pol κ at the 90 minutes time point does not decrease the total amount of DNA synthesis. We can add this data redundantly for the experiment shown in Figure 3C to satisfy the point raised by the referee.

Point 7 : We do not agree that the observation that Pol κ is required for the synthesis/stabilization of short replication intermediates is in discrepancy with the findings of Van et al., (JCB. 2010) since depletion of neither Pol δ nor Pol κ results in a complete disappearance of the short fragments (see Fig 5C-lane Δ Pol δ /90 min in Van et al. and Fig 3C from our manuscript, lane 90 min), nor in a total decrease of P-Chk1 (see Fig 6B from Van et al and Fig 1A-B from our manuscript). Our observations show that, besides Pol δ , another polymerase is implicated at replication forks stalled after aphidicolin treatment, and underline the complexity of the system. Moreover, in Supplementary Figure 2A and Figure 3E of our manuscript we have shown that Pol κ binds to chromatin in S-phase **only** in the presence of aphidicolin, which demonstrates that loading of Pol κ occurs after that of Pol δ since Pol δ is already bound to chromatin in S-phase. We would like to stress that the aim of our work has been to show a novel role for translesion Pol κ in the synthesis/stabilization of short replication intermediates formed at replication forks arrested with aphidicolin, and not to study the epistasis of DNA polymerases. We believe that the data provided in Figure 2 and 3 are robust enough to demonstrate a role for Pol κ in checkpoint activation in both mammalian cells and *Xenopus*.

Again we do not understand the single and double depletion experiment proposed by this referee. He/ She failed to specify what it is expected to be reduced by a first, and then a second round of depletion of Pol κ . We believe that this experiment is meaningless since a first round of depletion would leave enough Pol κ to fulfil its checkpoint function since Pol κ is in large excess in the *Xenopus* egg extract as most of other proteins stored in the egg.

Point 8. The possibility that removal of Pol κ would remove a factor essential for Pol δ DNA synthesis is very unlikely and not supported by the experiment shown in Figure 2D which demonstrates that the rate of DNA synthesis does not decrease after removal of Pol κ from egg extracts, which is expected to be the case if the polymerase activity of Pol δ would be affected.

Point 9a. We do not understand this point. We have stated in the paper (discussion page 18) that the interpretation of this experiment is that Pol κ may be required for replication forks restart. Also it has to be pointed out that in Figure 5 we monitored the number of BrdU-positive cells and not the intensity of the signal that would have given perhaps some information about the amount of ssDNA generated, although this assay is not quantitative nor enough sensitive to detect differences in short tracts of single stranded DNA.

Point 9b. We do not understand this comment either. PCNA monoubiquitylation is not required for checkpoint activation but depends upon replication fork uncoupling. In the presence of HU replication fork uncoupling occurs efficiently even in the absence of Pol κ (Supplementary Figure S3B and indeed Figure 4C) this is why we observe PCNA monoubiquitylation in the absence of Pol κ + HU (Figure 4C).

Point 10. We can perform a western blot with Cdc45 and Pol α antibodies to satisfy the point of this reviewer.

2nd Editorial Decision

18 January 2013

Thank you again for sending your responses and comments on the reviews for your recent submission, EMBOJ-2012-83985. I have now had a chance to look into your arguments, and I appreciate that many of them appear to be well taken. I did notice that referee 1 was considerably more negative than the other two, although I should add that during our routine pre-decision referee cross-commenting, referee 3 admitted that some of referee 1 points were reasonable and/or possibly caused by the manuscript not being sufficiently clear in various places - something that could hopefully be easily improved.

Importantly, I agree that repeating the experiments of Van et al would appear to make little sense, especially in light of your argumentations, while the alternative experiments you propose to query the pol kappa-delta relationship sound very promising. I would therefore be willing to considering a revised version of the manuscript once more for a single additional round of review, although I of course cannot guarantee the outcome of such re-assessment. Such a new manuscript should carefully take into account/respond to all the referees' points, and it would go back to at least referees 2 and 3 (the former actually being an expert in the *Xenopus* extract replication system), and possibly also to an additional arbitrating referee.

You will find below a hyperlink allowing you to upload a new manuscript and response letter once you are finished revising the study. Should you have any additional questions in the meantime, please do not hesitate to get in touch with me.

1st Revision - authors' response

02 May 2013

Reviewer #1

We do not understand the meaning of most of the points raised by this referee (points 2, 3, 5, 6, 9a and 9b), and the other remaining points are very arguable and not justified (7,8), nevertheless we have made an effort to satisfy some of them (points 1, 4, 6, 7, 10).

1) Fig1. The authors fail to provide details on how many times the experiments have been performed (especially in Figure 1c, which includes the rescue). Also, they did not monitor the level of proteins involved in checkpoint activation following siRNA treatment.

The experiments shown throughout the paper are representative examples of independent observations made several times and by different researchers. As required by the reviewer, we checked the level of major proteins involved in the checkpoint after Pol κ depletion with two independent siRNA in two independent cell lines. As shown in the new Fig S1E of the revised version, the level of ATR, ATRIP, CHK1, RAD9, HUS1, and POL δ is not affected after Pol κ depletion.

2) Endogenous levels of Pol K should be measured and compared to Pol alpha. It is indeed likely that the proposed role for Pol K is instead played by Pol alpha, whose concentration in egg extract is likely to be higher than Pol K

We do not understand at all this point. Even if the concentration of Pol α would be higher than Pol κ , we removed only Pol κ and we demonstrate in Figure 2C that in these conditions replication occurs very efficiently. Since Pol α is absolutely required for replication, this observation rules out the possibility that removal of Pol κ also removes Pol α . Moreover, we now show by immunoprecipitation experiments (Fig. S3H) that Pol κ does not interact with Pol α , and that depletion of Pol κ does not alter the level of neither Pol δ nor Pol α in both xenopus (Fig. S3E) and human cells (Fig. S1E), as expected.

3) Figure 2B: A control with p-Chk1 after UV is missing.

The level of Chk1 phosphorylation after UV irradiation is comparable to that obtained with aphidicolin as previously shown by us and others (Recolin et al., 2012; Nucl. Acids Res.; Byun et al., 2005; Genes & Dev.; Lupardus et al., 2000 Genes & Dev.). Since we present a better Fig 2B, this figure has been moved to supplementary information (new Fig. S3B) and we have added the requested control.

4) Figures 2B/2C: the loading control must be total Chk1.

We have now included Chk1 as a loading control as requested; furthermore, we showed that Chk1 is not co-depleted with Pol κ (see Fig 3A).

5) Figure 2D does not prove that depletion of Pol K does not affect DNA replication as it is claimed in the text. Δ mock and Δ PolK must be on the same graph with the same scale. If the scale is the same between the two graphs shown, the authors need to provide an explanation for the observation that Pol kappa depleted extract replicates ~20% better than mock. Also the graph lacks error bars.

We do not understand this point either. The new Figure 2C (previous Fig. 2D) clearly illustrates that the scale between the two graphs is identical. This experiment also shows that removal of Pol κ does not inhibit DNA synthesis and demonstrates that indeed Pol κ is not required for chromosomal DNA synthesis in the absence of aphidicolin, and this is also true in mammalian cells. We do reproducibly observe slightly more DNA synthesis in Pol κ -depleted extracts, which is very likely due to failure to activate the intra S phase checkpoint that restrains activation of late clusters of replication origins (Shechter et al., 2004. Nat. Cell. Biol.). This is entirely consistent with the observation that caffeine, an inhibitor of ATM/ATR, has no effect on DNA synthesis after removal of Pol κ (Fig. 2C).

As for the request of error bars, these are not used in measuring replication kinetics in *Xenopus* egg extracts. *Xenopus* egg extracts are very efficient in performing DNA replication *in vitro*, however the kinetics of DNA synthesis are variable from one batch of extracts to another. This depends upon the quality of eggs laid from one animal to another. Replicate of experiments are routinely performed using extracts from different batches, because it is often difficult to obtain large amount of a good extract from one batch of eggs laid by a single female. Moreover, it is a good practice to reproduce the same result using a different batch of egg extract. This is the main reason why error bars are not used on replication kinetics in *Xenopus* egg extracts, and this can be observed in all publications of established laboratories that make use of *Xenopus* eggs extracts (such as J. Blow, J. Walter and M. Méchali, and in recent publications of Dr. Maiorano's laboratory).

6) Figure 3C. As control, the authors should provide a replication assay of the experiment to ensure that the total incorporated dCTP was similar between Mock and depleted extract.

We have added this information (total dCTP incorporation) for the experiment shown in Figure 3C to satisfy the referee. This data shows that depletion of XPol κ results in even higher dCTP incorporation in the absence of aphidicolin, similar to what shown in Figure 2C, but this is significantly lower in the presence of aphidicolin, as expected if synthesis of small replication intermediates is inhibited.

7) The authors' model about the relationship between Pol delta and Pol K, regarding the generation of the short DNA fragments, which are thought to activate the checkpoint, is not supported by their results. Van et al showed that the generation of short DNA fragments is almost completely dependent on Pol delta, while the authors show that it is almost completely dependent on Pol K. This discrepancy should be experimentally addressed.

The authors should repeat the Pol delta experiment and verify the role of Pol delta in checkpoint activation in their conditions to confirm or disprove their hypothesis. This is a crucial point that needs to be addressed. Also, an epistasis analysis is required to prove that Pol delta and Pol kappa are on the same/or parallel pathways. Finally, If each polymerase is acting on a specific strand (as the authors propose), I would expect a partial reduction in the single depletion and a total reduction in the double depletion. This experiment should be performed to prove their point.

This is the major point of the reviewer #1 who raises the relationship between Pol κ and the replicative Pol δ regarding the generation of the short fragments at stalled replication forks after aphidicolin, important for the checkpoint activation. He/She pointed out the apparent discrepancy with the work of Van et al., (JCB. 2010) which showed the requirement for Pol δ in the synthesis of these small replication intermediates. In particular the reviewer argues that "Van et al. showed that the generation of short fragments is almost completely dependent on Pol δ , while the authors show that it is almost completely

dependent on Pol κ ". This argument is not justified, since depletion of neither Pol δ nor Pol κ results in a complete disappearance of the short fragments (see Fig 5C-lane Δ Pol δ /90 min in Van et al. and Fig 3C from our manuscript, lane 90 min), nor in a total decrease of P-Chk1 (see Fig 6B from Van et al and Fig 1A-B from our manuscript). Our observations show that, besides Pol δ , another polymerase is implicated at replication forks stalled after aphidicolin treatment, and underline the complexity of the system. According to us, there is no discrepancy between the two works. Moreover, as also stated in our manuscript, it has to be remembered that this finding is in agreement with the observation that, in contrast to the three replicative DNA polymerases Pol δ , Pol ϵ , and Pol α , TLS Pol κ is not inhibited by aphidicolin, and therefore is one DNA polymerase active at arrested forks. We do not exclude that both Pol δ and Pol κ act in a mutually dependent and coordinative manner. As we already mentioned to the editor, and he agreed with that, we do not think that repeating the experiment described in Van et al., consisting in removing Pol δ from egg extracts is justified and will be very informative. The aim of our work has been to demonstrate a novel role for a translesion DNA polymerase (Pol κ) in checkpoint activation, which is a completely novel concept in the field, and not to study the epistasis of this polymerase with respect to the replicative DNA polymerases. We believe that the data provided in Figure 2 and 3 are robust enough to demonstrate a role for Pol κ in checkpoint activation in both mammalian cells and *Xenopus*.

Nevertheless, we have performed a couple of experiments that give further insights into the relationships between Pol δ and Pol κ :

- We first investigated the binding of Pol α , δ , κ , PCNA and Rad9 to chromatin during DNA replication in *Xenopus* egg extracts, in the absence of aphidicolin or in the presence of two concentrations of aphidicolin, a low dose (15 μ M) which does not inhibit Pol α and a high dose (750 μ M) that prevents the primer synthesis by Pol α (Waga and Stillman, 1994 Nature; Michael et al., 2000 Science; see the new Fig 3E). The data clearly show that in absence of aphidicolin, Pol α , Pol δ , and PCNA are loaded onto chromatin as expected while Pol κ is not. In the presence of 15 μ M aphidicolin, Pol κ is recruited as well as Rad9, further supporting the requirement of Pol κ for the replication checkpoint upon replication stress. In contrast, in the presence of high dose of aphidicolin that inhibits primer synthesis, we found that only Pol α is present while PCNA, Pol δ and Rad 9 are not recruited as expected (Michael et al., 2000 Science; Maiorano et al., 2005 Cell; Byun et al., 2005 Genes & Dev.). In these conditions Pol κ is not recruited either suggesting that its loading onto chromatin is dependent on the primer synthesis by Pol α and/or pol delta upon replication stress.
- We also carried out a second set of experiments to investigate whether there is a physical interaction between Pol κ and Pol δ . We performed a series of immunoprecipitation experiments with Pol δ , Pol κ and Rad9 antibodies (see the new Figures Fig. S3G-H). The data clearly indicate that Pol κ interacts with Rad 9 (we provide new data of co-immunoprecipitation after XPolk and XRad9 immunoprecipitation experiments) but does not interact neither with Pol δ , nor with Pol α .

One scenario that could accommodate all these novel findings is that Pol κ binds to chromatin in S-phase only in the presence of replication stress (see Fig. 3E, *Xenopus* extracts and Fig. S2A, Human cells), and

that loading of Pol κ occurs after that of Pol α and/or Pol δ since they are already bound to chromatin in S-phase. Upon induction of replication stress DNA primase-Pol α is recruited to the leading strand to synthesize primers which are elongated by Pol κ in a distributive, rather than a processive manner by a replicative polymerase. It cannot be excluded that Pol κ may also function on the lagging strand in competition with DNA Pol δ . This regulation may ensure generation of multiple 5' ended primer-template junctions to enable efficient replication checkpoint activation and is in line with the observation that DNA synthesis at arrested forks continues at a slow rate, leading to the formation of small replication intermediates that contribute strongly to checkpoint activation. We have presented also evidence that in the absence of Pol κ the DNA binding of the Rad9 subunit of the 9-1-1 complex is affected after replication stress. These observations suggest that Pol κ -dependent DNA synthesis may be required for Rad9 binding. This interpretation is consistent with the observation that in the absence of Pol κ short replication intermediates are poorly made, resulting in much less 5' ends and consequently reduced recruitment of the 9-1-1 complex. Generation of multiple short DNA intermediates by Pol κ may also ensure that replication restart occurs as we have observed that removal of Pol κ from human cells leads to persistence of single-stranded DNA after recovery from a block with HU.

8) Is it possible that Pol K depletion removes some factors required for pol delta-dependent DNA synthesis? This should be discussed.

The possibility that removal of Pol κ would remove a factor essential for Pol δ DNA synthesis is very unlikely and not supported by the experiment shown in Figure 2C which demonstrates that the rate of DNA synthesis does not decrease after removal of Pol κ from egg extracts, which is expected to be the case if the polymerase activity of Pol δ would be affected. Furthermore, depletion of Pol κ in both *Xenopus* (New Fig. S3E) and human cells (New Fig. S1E) does not affect the level of Pol δ p125 and p66 subunits.

9a) I struggle to understand the interpretation of the ssDNA experiment (Figures 4A/4B). If Pol K is required for DNA synthesis in the presence of HU, then I would expect more ssDNA in the Pol K siRNA at the R0 point. Also, if Pol K siRNA also causes a checkpoint activation defect, then I would expect even more ssDNA, given that without the checkpoint, silent origins are free to fire. The result at the R3 point shows that ssDNA persists slightly longer after removal of HU, which suggests that the absence of Pol K affects replication fork restart, rather than DNA synthesis DNA during the HU/aphidicolin arrest.

We do not understand this point. We have stated in the paper (discussion page 19, lines 439-442) that the interpretation of this experiment is that indeed Pol κ may be required for replication fork restart. Also it has to be pointed out that in Figure 4 we monitored the number of BrdU-positive cells and not the intensity of the signal that would have given perhaps some information about the amount of ssDNA generated, although this assay is not quantitative nor enough sensitive to detect differences in short tracts of single stranded DNA.

9b) The authors claim that Pol K's function is required for checkpoint activation and that ubiquitinated PCNA is enriched on chromatin in these conditions. This is confusing given that others (Chang et al, 2006, cited in the text) demonstrated that checkpoint activation is independent of PCNA ubiquitination.

We do not understand this comment either. PCNA monoubiquitylation is not required for checkpoint activation but depends upon replication fork uncoupling (Chang et al., 2006). Upon replication stress, replication fork uncoupling occurs efficiently even in the absence of Pol κ as monitored by RPA loading

onto chromatin (Fig. 3B and Fig. S3C). The assay showed in Fig. 4C monitors the monoubiquitylated form of PCNA (Ub-PCNA), which is generated in response to replication stress by the ubiquitin E2-E3 complex composed of Rad6 and Rad18 and depends upon replication fork uncoupling. Higher amounts of Ub-PCNA bound to chromatin found in Pol κ -depleted cells after HU confirms that uncoupling occurred even in the absence of Pol κ .

10) Figure 5B. To ensure that the additional recruitment of RPA is due to ssDNA accumulation at stalled forks, the authors should verify that the recruitment of replication factors such as Cdc45 or Pol alpha is similar between control and Pol kappa siRNA treated cells.

We thank the reviewer for this interesting control. We have performed a new experiment to satisfy the point of this reviewer (see new Fig. S4A). This additional experiment made with two independent pol κ -siRNAs clearly shows that the level of Pol α and Cdc45 onto chromatin is similar in both control and Pol κ -depleted cells, indicating that RPA is recruited at ssDNA formed at stalled fork and not at new origins of replication.

Referee #2 (Remarks to the Author):

Betous and colleagues use siRNA in human tc cells and immunodepletion in frog egg extracts to study the role of PolK in activation of the replication checkpoint. The data show that loss of PolK compromises checkpoint activation, as measured by Chk1 phosphorylation. Follow-up experiments in frog egg extracts show that primer synthesis during a replication stress response is attenuated upon removal of PolK. Taken together, these data provide evidence for a role for PolK in checkpoint activation through primer extension, and as such they are worthy of publication once the minor issues detailed below have been taken care of.

Minor points

- 1) Introduction, 2nd paragraph. The authors state: "Independently, an alternative RFC-containing Rad17 complex recruits the heterotrimeric 9-1-1 complex to primers synthesized by DNA polymerase alpha, which in turn recruits the ATR activator TopBP1". The statement suggests a linear recruitment pathway where 9-1-1 loads first and then recruits TopBP1. Unfortunately, this statement is frequently made when summarizing checkpoint complex assembly, and there is little if any available data that support it. By contrast, two papers have shown that TopBP1 is required to recruit the 9-1-1 complex to sites of replication stress (Yan and Michael, JCB 184, p. 793, 2009 and Gong et al, Mol. Cell 37, p.438, 2010). I suggest that authors revise this sentence to reflect the currently available data.

We agree with this point. The reviewer is right, there is no strong data showing that 9-1-1 loads first and then recruits TopBP1. TopBP1 interacts with both the 9-1-1 and ATR-ATRIP complexes, and it plays a role in loading and/or stabilizing the 9-1-1 complex on damaged chromatin. We have revised this sentence in the text and we have incorporated additional references, including those indicated by the reviewer (see lines 80-83 page 4).

- 2) In Figure 2D DNA replication data are presented and plotted as "% of replication" over time. What does "% of replication" mean? How can we compare the replication efficiency of the two extracts (mock- and PolK-depleted)? This should be clarified.
In Fig 2C (previous Fig 2D), the "%" of replication means the amount of DNA replicated in the assay compared to the total amount of DNA present in the reaction at t=0 (input DNA). We have now clarified this point in the figure legend. This is the most rigorous measurement of the efficiency of replication in Xenopus egg extracts, and is crucial to interpret the data since in this system checkpoint activation relies upon nuclear membrane formation and replication only occurs when nuclear membrane has formed. The replication efficiency of the two extracts can be compared if the scale on the y axis is the same, which is the case in Figure 2C. Moreover, replication of both Mock and Pol κ-depleted extracts was performed at the same time but has been plotted into two separate graphs for clarity.
- 3) In Figure 3 a series of critical functional assays are performed with PolK-depleted extract, yet no control blots are shown that attest to the efficiency of depletion. Furthermore, no rescue of depleted extracts with recombinant PolK is shown for these experiments. These important controls should be included.

The efficiency of Pol κ depletion for the experiments shown in Figure 3 was already shown in panel A (Xpolk). The experiments described in panels A-D were performed from the same extract depleted of polk (panel A) from which chromatin and DNA samples were obtained. We found that addition of recombinant purified Pol κ rescued a functional replication checkpoint (Fig 2B) and we provide now a novel experiment shown in Fig. S3F showing that recruitment of Rad9 upon replication stress is also rescued by addition of purified recombinant Pol κ .

- 4) The blot in Figure 3D shows a ~50% reduction in the amount of Rad9 loaded onto aphidicolin-treated chromatin. Was this consistently observed? The authors should comment on this as the result has implications for leading vs lagging strand loading of 9-1-1 by PolK.

We reproduced this experiment three times and found reproducibly a reduction of about 50% in the amount of Rad9 after aphidicolin.

The contribution of lagging and leading strands to checkpoint activation is a very interesting question. To date, this issue is unclear. Available data from bacteria to *Xenopus* egg extracts are consistent with replication fork uncoupling at both lagging and leading strand after a replication block. Work from Cimprich's lab (Van et al., *J Cell Biol* 2010, 189, 233-246) has suggested that bulk synthesis of small nascent DNA occurs on the lagging strand, while on the leading strand the gap between nascent DNA primers is larger, as also observed in yeast (Lopes et al., *Mol Cell* 2006 21, 15-27). Therefore, it is possible that the two strands may contribute differently to checkpoint activation, since larger gaps on model templates have been shown to increase Chk1 phosphorylation (McDougall et al., *Genes & Dev.* 2007, 21: 898-903).

We thank the reviewer for pointing out that our result on Rad9 loading might have implication for leading versus lagging strand loading of the 9-1-1 complex by Pol κ . In contrast to Pol δ , Pol ϵ seems to be involved at a much lesser extent in the primer elongation process at an unwound fork after replication stress (Van et al., 2010), in agreement with the need of a distributive synthesis. The fact that Pol κ contributes to formation of replication intermediates at forks stalled with aphidicolin, together with the 50% reduction of Rad9 loading following Pol κ depletion, suggest that Pol κ may be directly involved in the elongation of these short DNA products on the leading strand.

- 5) The blot in Figure 5B uses actin as a loading control for the chromatin fraction (2nd panel down). This is an odd choice for a chromatin loading control and requires explanation.

We performed a novel experiment confirming that depletion of Pol κ leads to increased loading of RPA onto chromatin with a better chromatin loading control, MCM7 (see the new Fig S4A). In this new experiment, we also verified that the additional recruitment of RPA is due to ssDNA accumulation at stalled forks and not to firing of new replication origins, since we observed that the loading of initiation factors such as Cdc45 and Pol α remained unchanged following Pol κ depletion.

- 6) In Figure 6B the G2/M checkpoint is analyzed, but I see no indication that cells were induced to activate a checkpoint response (eg by HU or UV treatment). If these are indeed unperturbed cells then the experiment needs to be repeated with HU or UV to be meaningful.

We thank the reviewer for this important point. It is known that cell cycle arrest does not depend only on the Chk1-mediated pathway and there are at least two parallel pathways leading to G2/M arrest after external replication stress (Chk2 and p38). Indeed, abolition of G2/M arrest requires loss of both Chk1 and Chk2 function (see for example the work of Penny Jeggo's group in EMBO Journal 2006 (Stiff et al., EMBO J. 25, 5775-5782)). Moreover, in the absence of p53, cells depend on a third cell-cycle checkpoint pathway involving p38MAPK/MK2 for cell-cycle arrest and survival after DNA damage (see Reinhardt et al., Mol Cell 2007, 11, 175–189 from Michaël Yaffe's group). Consistent with this possibility, we now show in a new experiment (see below the additional Figure for the reviewer, not to be published) that Chk1 depletion by siRNA or reduction of Chk1 activation in Pol κ -depleted cells does not impact on the UV-induced G2/M arrest since Chk2 as well as p38 were normally activated in both cases.

G2-M checkpoint assay

(A) HeLa Cells were cultured and transfected with 50 nM of the indicated si-RNA as described in the manuscript. The sequence of Chk1 si-RNA are GAAGCAGUCGACAGUGAAGATT, UCUUCACUGCGACUGCUUCTT (Ahn et al, JBC 2003). 48h after transfection, cells were collected (R0) or irradiated with UV (20J/m²) and released for 7 hours (R7) in culture media with nocodazole (Sigma) at 0.1 μ M to capture cells that entered in mitosis. Cells were washed once in PBS, fixed in ethanol and permeabilized with 0.25% triton X-100. Mitotic cells were stained with H3P (Ser10, Cell signalling 1:75) as described (Betous et al, Mol Carcinog 2009). DNA was stained with Propidium iodide solution (25 μ g/mL Propidium iodide (Sigma) containing RNAse A (Sigma 0.1mg/ml). The mitotic index was measured by monitoring the phosphorylated form of Histone H3 by flow cytometry (FACSscan, Becton Dickinson). Results were analysed by using CellQuest software. The percentage of cells positive for P-Histone H3 is presented above each gate of the dot-plots.

(B) Half of the cell pellets used in (A) were lysed. Total cell lysates were subjected to SDS-PAGE electrophoresis followed by western blotting using the indicated antibodies.

Data shown in in Fig 7A and 7B indicate that in absence of external checkpoint activation, mild accumulation of cells in G2 and in M occurs in unperturbed Pol κ -depleted cells. Following the reviewer's comment, we have re-evaluated the interpretation of these data, since they cannot be explained by an endogenous G2/M defect. The observed slight cell accumulation in G2 and in M could result from delayed G2 and M phases due to the transmission of under-replicated DNA or unresolved replication intermediates in G2 and in M following Pol κ depletion. Indeed, we demonstrated that Pol κ is required to prevent transmission of incomplete replication intermediates through mitosis to G1 daughter cells (see Fig. 6A-C and Fig. S5B showing a significant increase in the number of spontaneous 53BP1 nuclear bodies in G1 daughter cells in the absence of Pol κ). We have incorporated this interpretation in the text p15-16 lines 365-368.

Referee #3 (Remarks to the Author):

This manuscript examines the role of the translesion DNA polymerase kappa in activation of the S phase checkpoint. A potential link has been suggested by the previously documented association of pol kappa with RAD9. There has been significant recent interest in the role of primer synthesis downstream of the replication fork stalls, thus far attributed to the action of pols alpha and delta. Here the authors suggest that pol kappa may also play a role in this primer synthesis and in the subsequent recruitment of the 9-1-1 checkpoint clamp, needed for efficient initiation of the S phase checkpoint. Thus, pol kappa is required for induction of Chk1 phosphorylation following arrest of replication with hydroxyurea and for recovery from replication stress. These are interesting and potentially significant observations. However, I feel a number of problems need to be addressed further.

1. The work appears to conflict with previously published data from Vaziri's group (Bi et al JBC 2005; Bi et al MCB 2006) who showed pol kappa-deficient MEFs exhibit a protracted S phase arrest following treatment with BDPE. However, this effect was not seen with HU. While the authors address this in the Discussion, the discrepancy remains rather inadequately investigated. Why would the proposed model not apply to both HU-induced arrest and to arrest caused by defective TLS? In other words it should not really make a difference to the proposed role for pol kappa whether the lesion causing the arrest also requires pol kappa for its replication. A possible resolution to these differences is the use of pol kappa deficient MEFs by Vaziri's group. It think it important to know how such cells behaved in the authors' experiments to know whether the difference simply reflects different cell types. Experiments on pol kappa κ -/- MEFs, which I believe have been produced by some of the present paper's authors, would also avoid any problems of incomplete pol kappa depletion.

This is obviously an interesting and important point and, as required by the reviewer, we now show, in the new panel F in Fig S1, a new experiment analysing the phosphorylation of Chk1 after HU treatment by using primary Pol κ -/- MEFs derived from polk κ -/- mice (Schenten et al., Eur. J. Immunol. 2002, 32, 3152-60). We clearly observed that there is no induction of P-Chk1 after HU treatment (in contrast to the data from Vaziri's group with the DNA damaging agent BPDE; see figure 7a from Bi et al., JBC 280, 22343, 2005). Moreover, we confirmed in mice that, in the absence of Pol κ , the phosphorylation of Chk1 is defective after HU, as we observed in human cells and in Xenopus extracts. As suggested by this referee, we requested the Pol κ κ -/-MEFs used in Bi et al., 2005, 2006 from Dr. C. Vaziri, but it seems that these cells are not available anymore. We also asked to Dr H. Ohmori in Japan who created the Pol κ nul mice, but we did not have any answer. So unfortunately we could not directly compare both MEF cell lines. We agree that the differences may well reflect different MEF cell types. This difference is also supported by the much higher UV sensitivity of our Pol κ -/- MEF cells that we used here (Schenten et al., Eur. J. Immunol. 2002, 32, 3152-60) compared to the MEF Pol κ κ -/- cells used by Vaziri's group which showed no significant sensitivity to UV light (see Fig. 6 from Bi et al., JBC 2005, 280, 22343). Furthermore, as pointed out by the reviewer (see point 3 below), the decreased level of Chk1 that we observed in our Pol κ κ -/- MEFs, which we think is due to a chronic replication stress induced by loss of Pol κ (see our

explanation below) is not observed in the MEFs from Vaziri's group (see figure 7a from Bi et al., JBC 280, 22343, 2005).

Collectively, these arguments suggest that the differences probably reflect different MEF cell types. We have incorporated these explanations in the discussion part, page 17, lines 392-399.

2. The authors propose a model in which pol kappa helps create a DNA substrate to which the 9-1-1 clamp can be recruited. How do the authors imagine pol kappa is recruited to perform this synthesis downstream of the arrested fork given that PCNA ubiquitination does not seem to be required for activation of the replication checkpoint.

Since we provide evidence for interaction between Pol κ and the Rad9 subunit of the 9-1-1 checkpoint clamp, our data may suggest that the major determinant of the recruitment of Pol κ is the binding of Rad9 at the 5' ends primer-template junctions accumulating through recycling of Pol α , either as a complex Pol κ -Rad9 or independently.

3. The diminution of CHK1 phosphorylation is rather variable between cell lines and in some cases the level of CHK1 itself appears to go down to. Why is this?

The experiments performed with extracts derived from activated *Xenopus* eggs synchronized in S-phase recapitulates *in vitro* the replication fork progression in S phase and the regulated activation of the replication checkpoint at stalled forks, and we never observed any decreased level of Chk1 in this system upon Pol κ depletion. In contrast, the diminution of the Chk1 level was observed in some cases in mammalian cells, and in particular in the Pol κ $-/-$ MEFs where the level of Chk1 is significantly affected upon Pol κ deficiency (see Fig S1F).

Cellular phenotypes result from multiple events occurring at all phases of the cell cycle and during complex signaling and cell cycle control pathways; the decreased level of Chk1 that we observed in some cellular context illustrates this cellular complexity. In the persistent absence of Pol κ , such in Pol κ $-/-$ MEFs, this is possibly due to a mild but chronic replication stress induced by loss of Pol κ , triggering prolonged fork stalling which can collapse and generate DSB, which in turn can activate slightly but chronically DNA damage checkpoint (see the enhanced γ -H2AX foci formation following Pol κ depletion in Fig. SF4), including the phosphorylation of Chk1 which is here independent of Pol κ . Since Chk1 activation by phosphorylation induces its degradation by the proteasome degradation system (Zhang et al., Mol Cell 2009, 35, 442–453), this can explain why Chk1 itself goes down. This is further evidenced by the ratio P-Chk1/Chk1 in untreated MEF Pol Kappa $-/-$ which is 1.67 higher compared to untreated control MEF. Interestingly, a recent JCB paper by Massimo Lopes (see Fig2 from Neelsen et al., J. Cell Biol 2013, 200, 699-708) showed that indeed a chronic replication stress by overexpression of cyclin E and Cdc25A triggers a fast decrease in Chk1 level. We discussed this point in the discussion (see page 17, lines 400-409).

4. In some blots, loading controls are missing (e.g. Fig 1D and E), while in others loading appears to be rather variable (e.g. Figs 2B and C).

We apologize for the absence or the quality of loading controls in the previous figures. We provide now better controls in the corresponding new Fig 1D (Actin as loading control), 1E (Chk1 as loading control) as well as 2B (Chk1 as loading control).

5. I don't find the difference in the number of RPA foci per cell between the control and pol kappa knockdown very convincing. What does this mean when compared with the total number of RPA positive cells?

We apologize for the lack of clear description. In figure 5A, left panel, the quantification of RPA-positive nuclei is presented. It indicates an increase of RPA-positive cells when Pol kappa is depleted (around 70% in Pol kappa depleted cells vs 20% in control cells). In the right panel of figure 5A we present the distribution of the RPA foci per nucleus. The position of the median is indicated and the p-value determined with the non-parametric Mann-Whitney test was 0.011. The difference is weak but significant. These results indicate that more cells contain ssDNA regions when Pol kappa is depleted and the amount of these ssDNA regions per cell is slightly but significantly increased.

6. Some of the blots are very cropped and it would be nice to see more, especially of CHK1. Are the phosphorylated forms of CHK1 visible and do they correlate with the phosphoCHK1 signal.

We apologize for the cropped blots of our previous version of the manuscript. We provide now in the revised version better Chk1 blots where no phosphorylated forms can be visible (see new Fig. 1D, 1E, S1F, S3B with non-cropped blots).

Thank you for submitting your revised manuscript on DNA Pol kappa and checkpoint activation for our consideration. It has now been assessed by the original referees 2 and 3, as well as by an additional fourth referee who was asked to comment on your responses to all three original reports. I am pleased to inform you that all three reviewers are largely satisfied by your responses and revisions. We should therefore be able to accept paper for publication, once the following minor/editorial points have been incorporated into the manuscript:

- Referees 3 and 4 request a few presentational modifications, concerning discussions and interpretations both in the text and in the final model figure; as well as an alteration to the title. Please modify the manuscript accordingly.

- Please add a brief 'Conflict of Interest' statement to the manuscript text, next to the acknowledgement and author contribution sections.

- Please combine all supplementary information (text and figures) into one single PDF of sufficiently high quality and sufficiently small size to facilitate download by our readers - keeping in mind that this part of the manuscript will not be type-set or copy-edited.

- Some blot/gel panels in Figure 3 appear to be assembled from discontinuous fragments. Please clarify and, if necessary, introduce visible separation lines between non-neighboring lanes as well as a brief justification/explanation in the figure legends.

- Finally, we encourage the publication of source data, particularly for electrophoretic gels and blots, with the aim of making primary data more accessible and transparent to the reader. I am taking this opportunity to invite you to provide a single PDF/JPG/GIF file per figure comprising the original, uncropped and unprocessed scans of the key gel/blot panels used in the respective figures. These should be labelled with the appropriate figure/panel number, and should have molecular weight markers; further annotation would clearly be useful but is not essential. A ZIP archive containing these individual files can be uploaded upon resubmission (selecting "Figure Source Data" as object type) and would be published online with the article as supplementary "Source Data" files associated with each relevant figure.

I am therefore returning the manuscript to you for one final round of minor revision, hoping you will be able to upload and re-submit the final corrected version as soon as possible. Should you have any questions in this regard, please do not hesitate to contact me directly.

REFEREE REPORTS:

Referee #2 (Remarks to the Author):

I have reviewed the rebuttal and revised MS and I feel that the authors have adequately addressed my concerns and thus I feel the paper should be accepted.

Referee #3 (Remarks to the Author):

The revisions that the authors present have largely clarified the points I raised in my review. A couple of points could be further improved in the text, however:

1. The authors have provided data from pol kappa-deficient MEFs and a potential explanation for the decrease in CHK1 levels based on the recent paper of Massimo Lopes. This is plausible, but not proven. While it is clearly not going to be possible to provide a direct comparison with the work of Vaziri et al., it would be helpful to record the strain background of the MEFs used in this paper in comparison with those of Vaziri, perhaps in the methods.

2. The discussion of the relationship between pol kappa and the 9-1-1 clamp is somewhat confusing still. On one hand the authors argue that pol-k dependent DNA synthesis is required for RAD9 loading (e.g. bottom of pg 18) but on the other hand also argue that the recruitment of pol kappa may be dependent on its interaction with RAD9. I appreciate that it is not really possible to really say which comes first, but I think a bit more clarity could be brought to this issue, perhaps by modifying Figure 8, which currently doesn't address the proposed relationship between priming by pol alpha, 9-1-1 loading and pol kappa recruitment.

The authors' response to the comments of reviewer 2 appear to have been rationally and reasonably addressed.

With regard to the concerns of reviewer 1, I am less convinced that that the data of Van et al and those in the current paper concerning the relationship between pol delta and pol kappa are really in such strong conflict (point 7). It seems clear that there is potential redundancy between the two polymerases. While a detailed examination of the epistasis between the two will be informative, I don't think it's reasonable to expect the current authors to essentially repeat all the experiments of Van et al. I think the other points have also been reasonably addressed. Concerning point 9, surely firing of silent origins would suppress persistent and long ssDNA by replication of these regions from the opposite direction?

Referee #4 (Remarks to the Author):

Betous, Pillaire and co-workers report on a wide-ranging set of investigations regarding a new function of the specialized vertebrate DNA polymerase POLK in regulating cellular responses to spontaneous and stress-induced DNA replication stalling. The work reveals new phenotypes, including spontaneous replication stress arising after suppression of POLK.

In the revised version, the authors have responded to the comments of all the referees by including additional experiments and more thorough explanations. The comments, including those of Referee 1, certainly helped to improve the paper in revision, by emphasizing the need for better quantification and statistical analysis of the data.

I believe that the title of the paper should be changed. Currently it is "DNA polymerase kappa - dependent DNA synthesis at stalled replication forks is required for checkpoint activation".

There are two problems with this. First, "required" is a very strong word, and although POLK seems important, it does not seem to be required. As shown in Fig 1 for mammalian cells, some CHK1 phosphorylation remains when polk is suppressed. In Fig 2B, 2D there is some checkpoint signaling remaining in xenopus extracts efficiently depleted of pol kappa. It is more accurate to say that Chk1 phosphorylation is invoked at a lower level following induced replication stalling in pol kappa-defective cells. Second, as discussed well in the paper, activation of some checkpoints still occurs in the absence of POLK (e.g. CHK2). The consequence is that, as described in response to point 6 of reviewer 2, there is no significant problem with G2/M arrest in POLK-defective cells.

I suggest this more accurate title: "DNA polymerase kappa -dependent DNA synthesis at stalled replication forks is important for CHK1 activation".

"Required" should not be used in line 288.

And probably not the word "necessary" in lines 298-299. Instead one can say that polk activity promotes efficient phosphorylation.

Note that Fig 2 part D is not labeled in my copy.

Please find below our point-by-point response to the minor comments of the referees #3 and #4. We have modified the manuscript accordingly concerning the title, discussions and interpretations both in the text and in the final model figure.

We have added a brief 'Conflict of Interest' statement to the manuscript text, next to the acknowledgements and author contribution sections.

We have combined all supplementary information (text and figures) into one single PDF of sufficiently high quality and sufficiently small size to facilitate download.

We have clarified some blot panels in Figure 3 with increased separation lines between non-neighboring lanes.

Finally, we have provided a single PDF files (one per figure) comprising original scans of the key blot panels, with appropriate figure and panel numbers and molecular weight markers.

We hope that you now find the manuscript to be suitable for publication in *The EMBO Journal*, and we look very much forward to be hearing from you soon.

Best regards,

Jean-Sébastien Hoffmann and Domenico Maiorano

Referee #3 (Remarks to the Author):

The revisions that the authors present have largely clarified the points I raised in my review. A couple of points could be further improved in the text, however:

1. The authors have provided data from pol kappa-deficient MEFs and a potential explanation for the decrease in CHK1 levels based on the recent paper of Massimo Lopes. This is plausible, but not proven. While it is clearly not going to be possible to provide a direct comparison with the work of Vaziri et al., it would be helpful to record the strain background of the MEFs used in this paper in comparison with those of Vaziri, perhaps in the methods.

The embryonic fibroblasts (MEF) used in this study were generated from homozygous mutant Pol κ -deficient mice described in Schenten et al. (2002) DNA polymerase kappa deficiency does not affect somatic hypermutation in mice. *Eur J Immunol* **32**: 3152-3160. We have included this information in the Supplemental Materials and Methods part.

2. The discussion of the relationship between pol kappa and the 9-1-1 clamp is somewhat confusing still. On one hand the authors argue that pol-k dependent DNA synthesis is required for RAD9 loading (e.g.

bottom of pg 18) but on the other hand also argue that the recruitment of pol kappa may be dependent on its interaction with RAD9. I appreciate that it is not really possible to really say which comes first, but I think a bit more clarity could be brought to this issue, perhaps by modifying Figure 8, which currently doesn't address the proposed relationship between priming by pol alpha, 9-1-1 loading and pol kappa recruitment.

We thank the reviewer for this suggestion. Since the presence of stable 5' ends of primer-template junctions constitutes the likely binding sites for the 9-1-1 complex (Ellison & Stillman, *PLoS biology* 2003), we essentially reasoned and privileged the hypothesis throughout the manuscript that Pol κ -dependent DNA synthesis is required for the 9-1-1 complex binding (see lines 243-246 and 268-271 in the results part and lines 434-439 in the discussion part). Therefore, for clarity, we decided to present in the final model in Figure 8 the sequential action on the leading strand of Pol α (RNA primers), Pol κ (generation of short DNA fragments) and the loading of the 9-1-1 complex.

Referee #4 (Remarks to the Author):

1. I believe that the title of the paper should be changed. Currently it is "DNA polymerase kappa - dependent DNA synthesis at stalled replication forks is required for checkpoint activation". There are two problems with this. First, "required" is a very strong word, and although POLK seems important, it does not seem to be required. As shown in Fig 1 for mammalian cells, some CHK1 phosphorylation remains when polk is suppressed. In Fig 2B, 2D there is some checkpoint signaling remaining in xenopus extracts efficiently depleted of pol kappa. It is more accurate to say that Chk1 phosphorylation is invoked at a lower level following induced replication stalling in pol kappa-defective cells. Second, as discussed well in the paper, activation of some checkpoints still occurs in the absence of POLK (e.g. CHK2). The consequence is that, as described in response to point 6 of reviewer 2, there is no significant problem with G2/M arrest in POLK-defective cells. I suggest this more accurate title: "DNA polymerase kappa - dependent DNA synthesis at stalled replication forks is important for CHK1 activation".

We have changed the title as suggest by the reviewer to "DNA polymerase kappa - dependent DNA synthesis at stalled replication forks is important for Chk1 activation"

2. "Required" should not be used in line 288.

We changed "required" to "important" in line 288.

3. And probably not the word "necessary" in lines 298-299. Instead one can say that polk activity promotes efficient phosphorylation.

As proposed by the reviewer, we changed the sentence: "Pol κ activity promotes efficient phosphorylation of Chk1" in lines 298-299.

4. Note that Fig 2 part D is not labeled in my copy.

We apologize for the lack of clarity of the legend of this figure 2D which represents two panels of the kinetics of DNA synthesis of mock-depleted (Δ Mock; left panel) or XPol κ -depleted (Δ XPol κ ; right panel) egg extracts. We have incorporated this precision in the legend of the Figure 2.

Thank you for submitting your final revised manuscript for our consideration. I am pleased to inform you that we have now accepted it for publication in *The EMBO Journal*.

Thank you again for this contribution to *The EMBO Journal* and congratulations on a successful publication! Please consider us again in the future for your most exciting work.